# Functional development of the human cerebellum from birth to age five

Wenjiao Lyu [1,2,3], Kim-Han Thung[1,2,3], Khoi Minh Huynh [1,2,3], Li Wang [1,2], Weili Lin [1,2], Sahar Ahmad [1,2] & Pew-Thian Yap [1,2] ✉

Despite the cerebellum's crucial role in brain function, its early development, particularly in relation to the cerebrum, remains poorly understood. Here, we examine cerebellocortical connectivity using over 1000 high-quality resting-state functional MRI scans from children between birth and five years of age. By mapping cerebellar topography with fine temporal granularity, we unveil the hierarchical organization of cerebellocortical functional connectivity from infancy. We observe dynamic shifts in cerebellar functional topography, which become more focal with age while largely maintaining stable anchor regions similar to adults, highlighting the cerebellum's evolving yet organized role in functional integration during early development. Our findings demonstrate cerebellar connectivity to higher-order networks at birth, which generally strengthen with age, emphasizing the cerebellum's early role in cognitive processing beyond sensory and motor functions. Our study provides insights into early cerebellocortical interactions, reveals functional asymmetry and sex-specific patterns in cerebellar development, and lays the groundwork for future research on cerebellum-related disorders in children.

The cerebellum's contributions to human language, emotional regulation, attention control, cognition, and working memory, beyond its widely acknowledged role in motor functions, have garnered increasing attention in recent years[1–3]. These diverse functions are believed to result from the cerebellum's interaction with the cerebrum, positioning the cerebellocortical system as one of the most crucial circuits in the human brain[4]. Evolutionary anthropologists suggest that brain expansion in primates, including humans, is driven by the selective modular expansion of the cerebellocortical system[5]. Notably, compared to other primates, key regions associated with cognition in the human cerebellocortical system expand at the highest evolutionary rates[5,6], indicating that this system may play a pivotal role in the evolution of human brain functions and may offer critical clues on the formation of higher-order cognitive functions in humans. A growing body of evidence from both non-human primate experiments and human neuroimaging studies has confirmed the existence of the connections between the cerebellum and the cortex[7]. Additionally, clinical studies[8,9] have shown that damage to cerebellocortical

connections during development can lead to both cerebellar abnormalities and disrupted cortical development, highlighting the critical role of cerebellocortical connectivity in early neurodevelopment. However, there is still limited knowledge about typical cerebellocortical connectivity and how it develops in early human life.

Children progress through several distinct stages in early childhood—neonatal, infancy, toddlerhood, and preschool—each marked by rapid motor and cognitive development. In the neonatal stage, children develop basic reflexes and motor responses, such as grasping, head movements, and recognizing their mother's voice. As they transition into infancy, they begin to acquire more complex motor skills, like rolling, sitting, and standing, while gradually displaying higher-order functions like recognizing familiar faces and objects, using simple gestures, and beginning to speak. The toddler stage sees children mastering walking, running, jumping, and throwing, which require coordination, balance, and motor planning. Additionally, they also experience a wider range of emotions, imitate behaviors, and increasingly use language for communication[10–15]. During the

[1]Department of Radiology, University of North Carolina, Chapel Hill, NC, USA. [2]Biomedical Research Imaging Center, University of North Carolina, Chapel Hill, NC, USA. [3]These authors contributed equally: Wenjiao Lyu, Kim-Han Thung, Khoi Minh Huynh. ✉e-mail: ptyap@med.unc.edu

preschool years, children further refine these skills and begin to develop fine motor abilities, like drawing, as well as higher-order cognitive functions such as counting, categorization, language development, and storytelling[15,16]. The rapid development of these higher-order functions, particularly before school age, distinguishes humans from other mammals. This intense developmental period has prompted researchers to study how the cerebellum's unique connection to the cerebral cortex supports these advanced abilities.

The development of these motor and cognitive skills is supported by the maturation of neural substrates in the brain after birth, involving processes such as synaptogenesis (formation of new synapses), myelination (insulation of nerve fibers), programmed cell death (apoptosis), and synaptic pruning (removal of unnecessary synapses). These processes occur at different rates across various brain regions during early childhood[17,18]. For example, the visual cortex, which processes sensory input, matures earlier than the prefrontal cortex, which is associated with decision-making and higher cognitive functions[17]. This difference in developmental timing shows that sensory and association regions—key components of the brain's hierarchical organization—do not mature simultaneously. As a result, the organization and strength of cerebellocortical connections—the pathways linking the cerebellum and the cerebral cortex—are likely to differ significantly between children and adults. Understanding these differences can provide insights into how the brain supports the rapid development of complex sensorimotor and cognitive abilities during early childhood.

Over the past few decades, functional magnetic resonance imaging (fMRI) has been used to study the functional connectivity between the cerebellum and cortical networks. Advances in this field have revealed the macroscale functional organization of the adult cerebellum based on its connections with cortical regions. Key findings include the cerebellum's functional division into primary sensorimotor and supramodal zones[19]; its contralateral connectivity with the cerebral cortex[20]; the presence of multiple topographically organized somatomotor representations[20]; the involvement of cerebellar subregions in both integrative and segregative functions[21]; the contribution of phylogenetically recent regions, Crus I and Crus II, to connections with higher-order cognitive networks[22]; and a functional hierarchy along the central axis of motor and nonmotor organization in the cerebellum[23–25]. Additionally, detailed functional mapping of the cerebellum has been achieved at both the group[20] and individual[26] levels.

Despite advances in understanding cerebellocortical functional connectivity in adults, studies in young children remain limited[27,28]. This gap in knowledge is particularly concerning given that altered cerebellocortical functional connectivity has been observed in early-onset neurodevelopmental disorders such as autism spectrum disorder (ASD) and attention deficit hyperactivity disorder (ADHD)[29–34]. Elucidating typical cerebellocortical functional connectivity during early development is essential for establishing a foundational understanding that can inform the interpretation of these abnormalities and their broader implications on neurodevelopmental disorders. Several fundamental questions about early cerebellar development remain unanswered:

- When does the cerebellum first begin to contribute to higher-order functions?
- How do cerebellar functional patterns evolve during early development?
- At what age does children's cerebellar functional topography resemble that of adults?
- How does the early development of cerebellocortical connectivity differ between females and males?
- Is there a difference in functional cerebellar development between the left and right hemispheres, and if so, when does this lateralization commence?

- What is the spatiotemporal pattern of early cerebellocortical connectivity development?
- How does the functional hierarchy of the cerebellum evolve throughout early development?

To answer these questions, we mapped the functional connectivity patterns between the cerebellum and cerebral resting-state networks (RSNs) in children from birth to 60 months, using over 1000 fMRI scans from 275 participants in the Baby Connectome Project (BCP)[35]. We examined the development and organization of cerebellocortical functional connectivity over time, focusing on the establishment and timing of the cerebellar connectivity with higher-order networks. Additionally, we explored spatiotemporal patterns, lateralization, and sex differences in cerebellocortical functional connectivity, contributing to a broader understanding of the cerebellar functional architecture during the critical period of early development.

## Results

We applied group independent component analysis (GICA) to preprocessed fMRI data to generate 40 RSNs, from which we selected 30 cortical RSNs to ensure full coverage of the cerebral cortex. These 30 RSNs were categorized into eight large-scale networks: sensorimotor network (SMN), auditory network (AUD), visual network (VIS), salience network (SN), ventral attention network (VAN), default mode network (DMN), executive control network (ECN), and dorsal attention network (DAN) (Fig. 1). Each RSN was named based on its network affiliation and primary brain location (Table S1). To clarify the principles of cerebellocortical organization, we grouped the SMN, AUD, and VIS into primary networks, and the SN, VAN, DMN, ECN, and DAN into higher-order networks, depending on whether they are primarily engaged in sensory and motor functions or higher-order cognitive functions.

### Cerebellocortical functional connectivity

To validate our analytic framework, we first mapped the cerebellar representations of the foot, hand, and tongue to assess whether functional organization was accurately captured[20,26]. Accordingly, we evaluated the functional connectivity between the SM-Foot, SM-Hand, SM-Tongue, and the cerebellum and observed that sensorimotor representations in the cerebellum during early childhood exhibit organizational characteristics similar to those in adults. Specifically, in the anterior lobe, the representation is inverted, with the foot anterior to the hand and tongue, whereas in the posterior lobe, the representation is upright, with the tongue anterior to the hand and foot (Fig. 2a). Using the spatially unbiased atlas template (SUIT) software package developed by Diedrichsen and colleagues[36], we confirmed that the SMN representations are situated at expected positions on the flat map (Fig. 2b), gauging based on lobular dermacation (Fig. 2c). Subsequently, we depicted the spatial and temporal changes in cerebellocortical functional connectivity using network-specific flat maps (Fig. 3 and Fig. S2) and age-related trajectories (Fig. S3).

For primary networks (Fig. 3, primary networks), heightened connectivity was observed between several cerebellar regions, encompassing Lobules I–VI and VIII, and the SMN in early childhood, in line with previous studies on adults[20,26,37]. Specifically, Lobules I–V, the lateral aspect (away from the vermis) of Lobule VIII, and Vermis VI–VIII exhibit robust connectivity with SM-Foot. Lobules V–VI and the medial aspect (towards the vermis) of Lobule VIII exhibit strong connectivity with SM-Hand. As expected, the right cerebellum exhibits more pronounced connectivity with the left SM-Hand, and vice versa. Additionally, Lobule VI and the medial aspect of Lobule VIIIa exhibit strong connectivity with SM-Tongue. Interestingly, unlike previous studies focused on adults, connectivity between the cerebellum and AUD is evident in early childhood, particularly during the first few months of life (Fig. S2, primary networks). Additionally, we identified robust connectivity between the VIS and multiple cerebellar regions,

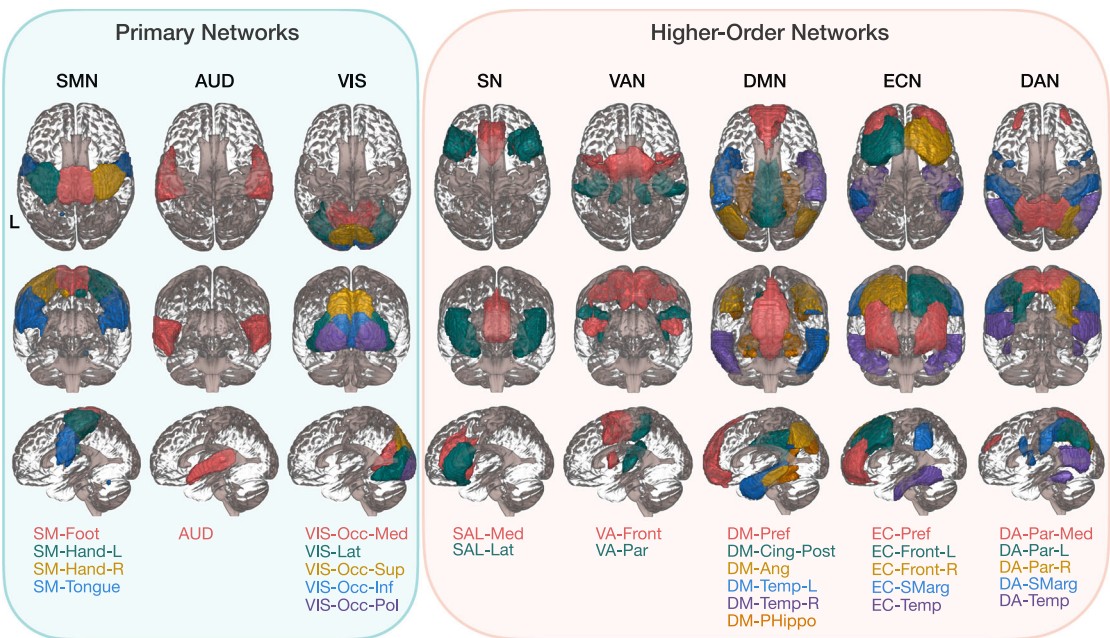

**Fig. 1 | Cortical networks.** Except the VIS and the DAN, which are presented from superior to inferior, posterior to anterior, and left to right, all networks are displayed from superior to inferior, anterior to posterior, and left to right. Resting-state networks in each large-scale network are color-coded to aid visualization. SMN sensorimotor network, AUD auditory network, VIS visual network, SN salience network, VAN ventral attention network, DMN default mode network, ECN executive control network, DAN dorsal attention network, SM sensorimotor, L left, R right, Occ occipital, Med medial, Lat lateral, Sup superior, Inf inferior, Pol pole, SAL salience, VA ventral attention, DM default mode, Pref prefrontal, Cing cingulate gyrus, Post posterior, Ang angular gyrus, Temp temporal, PHippo parahippocampus, EC executive control, Front frontal, SMarg supramarginal gyrus, DA dorsal attention, Par parietal.

including Lobule VI, Crus I, and the vermis, encompassing a broader spatial extent than previously reported in the literature[20,26,38]. Notably, cerebellar regions exhibiting strong connectivity to VIS shrink with age, mainly localizing at the Lobule VI and Vermis VI−VII from 48 months (Fig. S2, primary networks), potentially indicating network specialization refinement.

Consistent with prior research on adults, cerebellar regions exhibiting heightened connectivity with higher-order networks are predominantly located in Lobules VI, Crus I, Crus II, IX, and X (Fig. 3 and Fig. S2, higher-order networks). Despite the cerebellum's typically weak and diffuse connectivity with most higher-order networks, strong connections were observed between the cerebellum and certain higher-order RSNs such as DM-PHippo, DM-Pref, bilateral EC-Front, EC-Smarg, and EC-Temp, which were evident even at birth. Additionally, a trend of predominant contralateral connectivity was observed between the cerebellum and ECN-Front. Specifically, the right cerebellum is predominantly connected with left ECN-Front, whereas the left cerebellum is predominantly connected with right ECN-Front. The cerebellum generally exhibits stronger connectivity with primary networks compared to higher-order networks, especially during the first year of life, as demonstrated by the trajectories of the peak connectivity (Fig. S3a). Biweekly growth rates indicate that connectivity between the cerebellum and the SM-Hand-L (primarily associated with right-hand motor functions) and SM-Tongue (primarily associated with tongue motor functions) increases with age, while connectivity with most primary networks remains stable or declines. Conversely, connectivity between the cerebellum and most higher-order networks tends to increase with age (Fig. S3b). Together, these findings reveal dynamic changes in the spatial distribution and strength of cerebellar connectivity with RSNs during early childhood.

### Cerebellar functional topography

To better capture the spatiotemporal evolution of cerebellocortical functional connectivity across early childhood, we employed a winner-take-all approach to identify the network with the strongest connection to each cerebellar voxel[20,27,39,40], generating parcellation maps at three levels of granularity: (i) coarse granularity with two networks (primary and higher-order networks), (ii) medium granularity with eight networks (large-scale networks), and (iii) fine granularity with thirty networks (resting-state networks) (Fig. 4a).

At coarse granularity (Fig. 4a, top row), we observed that although the cerebellum is predominantly connected to primary networks during the first few months after birth, regions in the posterior cerebellar lobe−particularly Crus I, Crus II, Lobule VIIb, and Lobule VIII−as well as those in the flocculonodular lobe that connect to primary networks shrink over time. By 60 months, cerebellar regions linked to primary networks are primarily restricted to Lobules I−VI, Lobule VIII, and Vermis VI−VIII, exhibiting an adult-like pattern. In contrast, regions in the posterior and flocculonodular lobes assigned to higher-order networks expand with age. Given that primary networks align with cortical sensory regions, while higher-order networks align with association regions, these findings may suggest a developmental hierarchy stratified by the sensory-association (S-A) axis[41].

At medium granularity (Fig. 4a, middle row), regions in the anterior lobe connected to the SMN remain relatively stable throughout early childhood, while regions in the posterior and flocculonodular lobes connected to the SMN gradually shrink and concentrate in Lobule VIII by 60 months. Despite the presence of large cerebellar regions connected to the AUD in the first few months after birth, these regions notably decrease from 6 months onward, disappearing almost entirely before reappearing in small patches around 36 months. Initially, most cerebellar regions connected to the VIS are located in Lobule VI and Crus I. However, as age increases, the areas of Lobule VI and Crus I connected to VIS decrease, and by 60 months, the cerebellar regions connected to VIS are mainly concentrated at Vermis VI. Notably, Crus I, Crus II, and Lobule VIIb of the cerebellum are predominantly connected to the ECN, DMN, DAN, VAN, and SN−networks associated with higher-order functions−for most of early childhood (starting around 6 months).

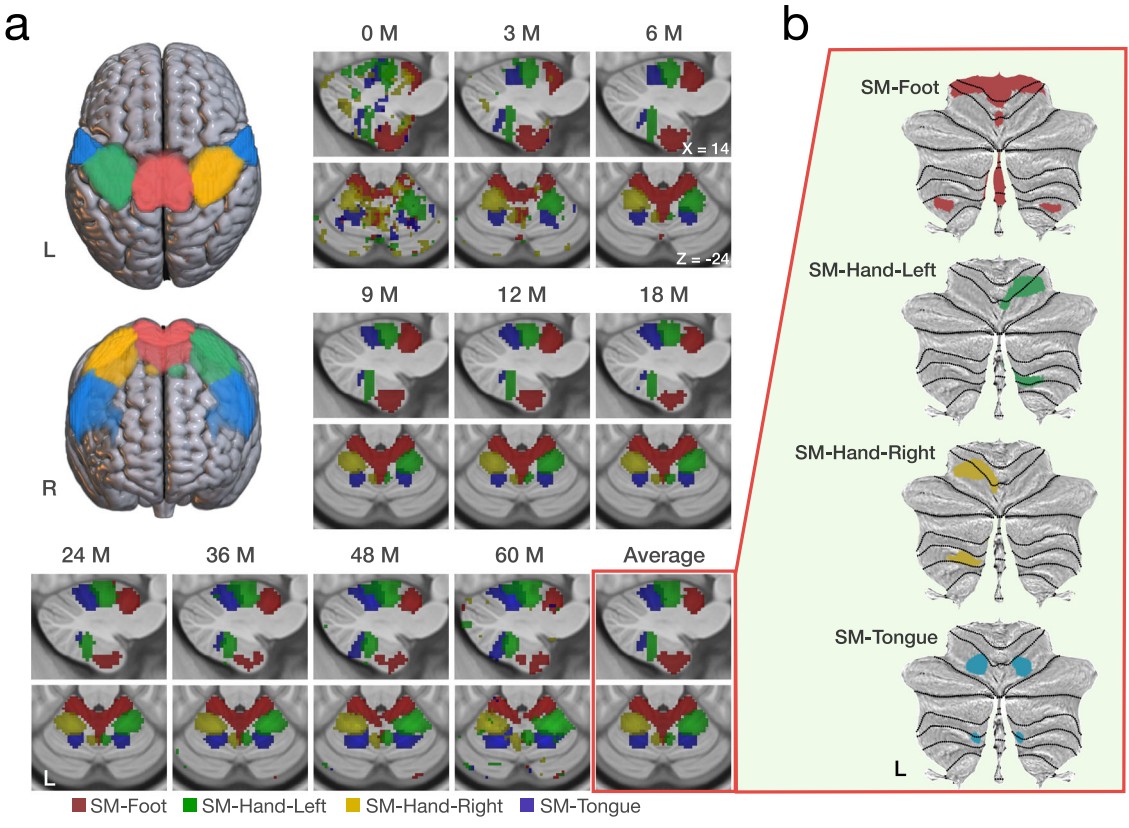

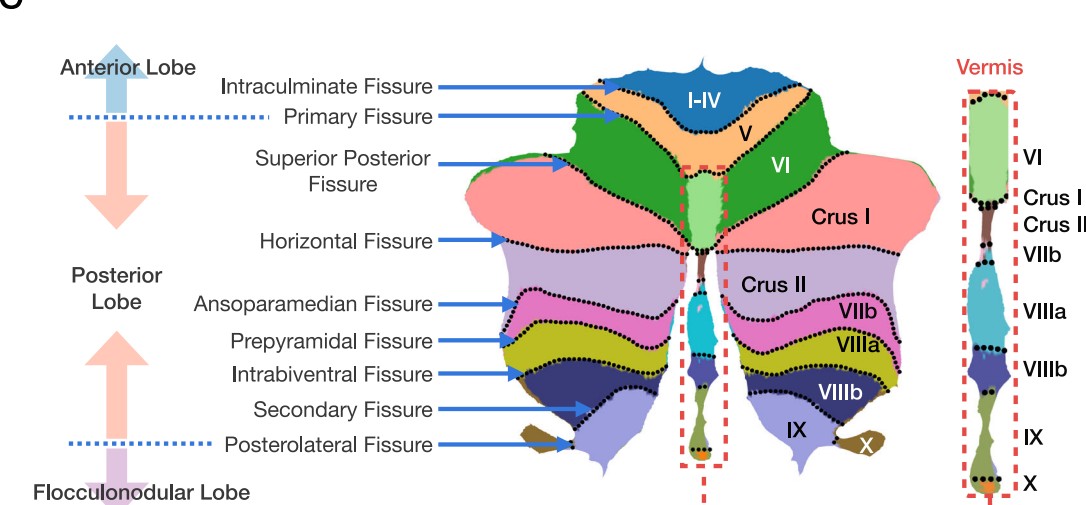

**Fig. 2 | Representations of sensorimotor networks. a** Sensorimotor networks and their representations in the cerebellum across age, along with the average sensorimotor representation in MNI space, thresholded at $z = 3.5$. **b** Flat maps of the average sensorimotor representations in the cerebellum, thresholded at $z = 3.5$. **c** Lobular, fissural, and lobar annotations on the cerebellar flat map[36,37,108].

Functional parcellation at fine granularity (Fig. 4a, bottom row) provides a more detailed depiction of the topography of each RSN in the cerebellum, capturing subtle spatiotemporal changes that might be overlooked at medium granularity. For example, the fine-granularity parcellation revealed characteristic contralateral cerebellar connectivity with cortical networks, which extends beyond the SMN-related RSNs. Specifically, the right cerebellum tends to connect to SM-Hand-L, DM-Temp-L, EC-Front-L, and DA-Par-L, whereas the left cerebellum tends to connect to SM-Hand-R and EC-Front-R. Notably, RSNs within the same large-scale network are often topographically adjacent in the cerebellum, even if they are spatially distant in the

cerebral cortex. For example, EC-Front, EC-Temp, and EC-SMarg, which belong to the same large-scale network, the ECN, are topographically non-adjacent in the cortex but are adjacent in the cerebellum at 60 months, highlighting the cerebellum's pivotal role in integrating brain function[27,42].

We also compared the cerebellar functional topography maps obtained from 60-month-old children with Buckner et al.'s cerebellar resting-state networks[43] and the multidomain task battery (MDTB) cerebellar parcellation maps of adults[44] (Fig. 4b). Note that the MDTB parcellation map, derived from a set of tasks[44], is labeled according to each region's most defining function, although the region may be

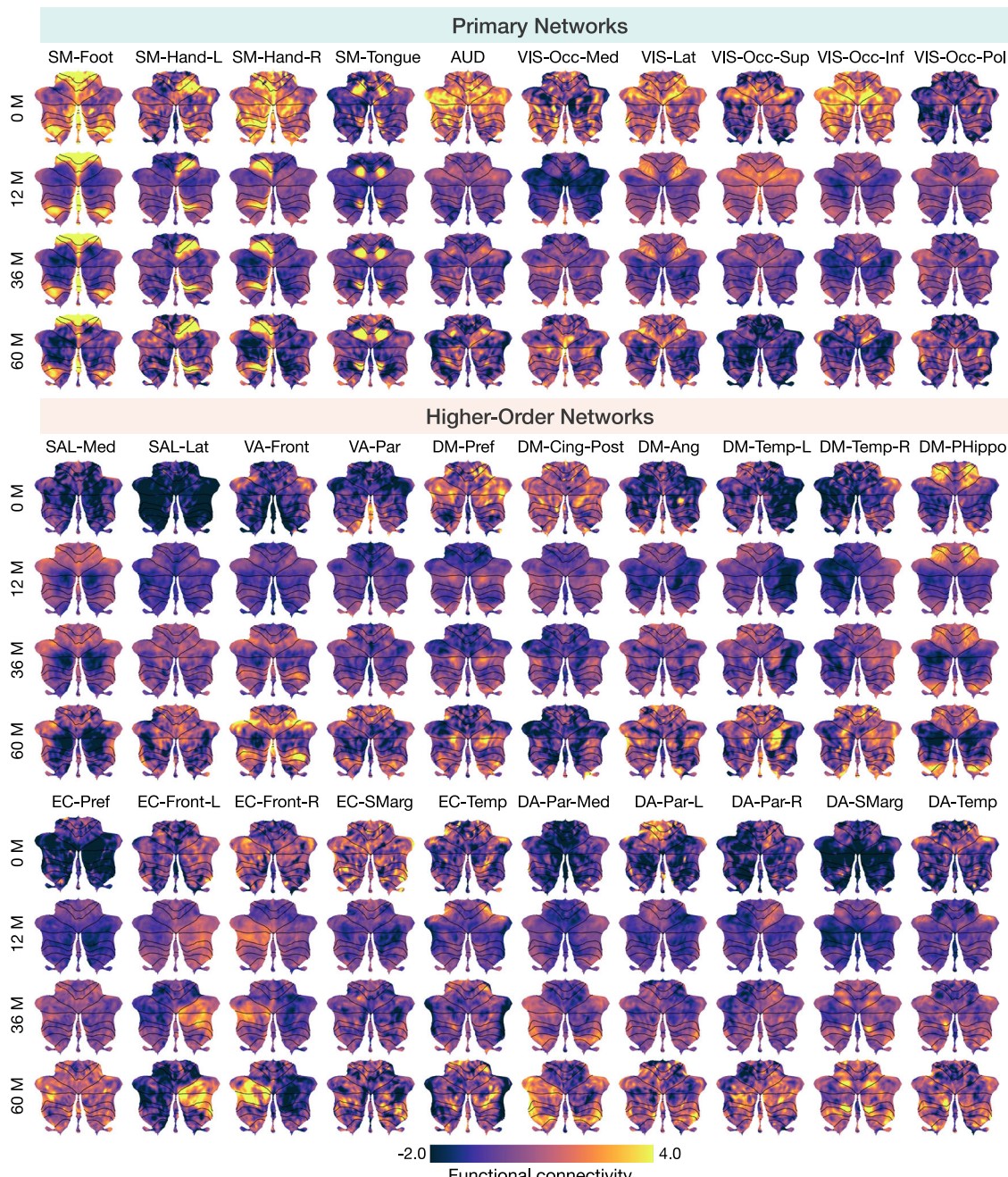

**Fig. 3 | Cerebellar functional flat maps.** Spatiotemporal patterns of connectivity (z-transformed) between the cerebellum and RSNs, grouped into primary networks and higher-order networks, at birth, 12 months, 36 months, and 60 months. Values outside the range of − 2.0 to 4.0 are capped for clarity.

associated with multiple functions. At 60 months, the cerebellar functional topography of primary and higher-order networks in children exhibits a distinct hierarchical organization (Fig. 4b, Children−2 Networks). Regions predominantly connected to primary networks are mainly situated within Lobules I–VI and the medial aspect of Lobule VIII, wheareas regions dominated by higher-order networks are primarily located in Lobule VII (including Crus I, Crus II, and Lobule VIIb), the lateral aspect of Lobule VIII, Lobule IX and Lobule X, closely resembling the functional organization principles of adults, particularly the "double motor and triple nonmotor representation"[23,24] or "three-fold organization" pattern[45].

From 36 to 60 months, the cerebellar functional topography of the eight large-scale networks in children (Fig. 4a, middle row) progressively mirrors that of the seven resting-state functional networks

in adults (Fig. 4b, Adults−7 Networks) with the following pattern of cerebellocortical functional connectivity:

- SMN − Lobules I–VI, Lobule VIII, and Vermis VIII.
- AUD − Small regions in Lobules VI and VIIb.
- VIS − Lobule VI and Vermis VI.
- SN − Lobule VI and Crus I.
- VAN − Lobule VI, Crus I, and Lobule VIII.
- DMN − Crus I, Crus II, Lobules IX–X, Vermis IX, and Vermis X.
- ECN − Crus I, Crus II, and Lobule VIIb.
- DAN − Crus I, Lobule VIIb, Lobule VIII, Lobule IX, and Lobule X.

In 60-month-old children (Fig. 4b, Children−8 Networks), the cerebellar connectivity topography resembles that of adults but has not yet fully matured. Compared to adults, children show greater

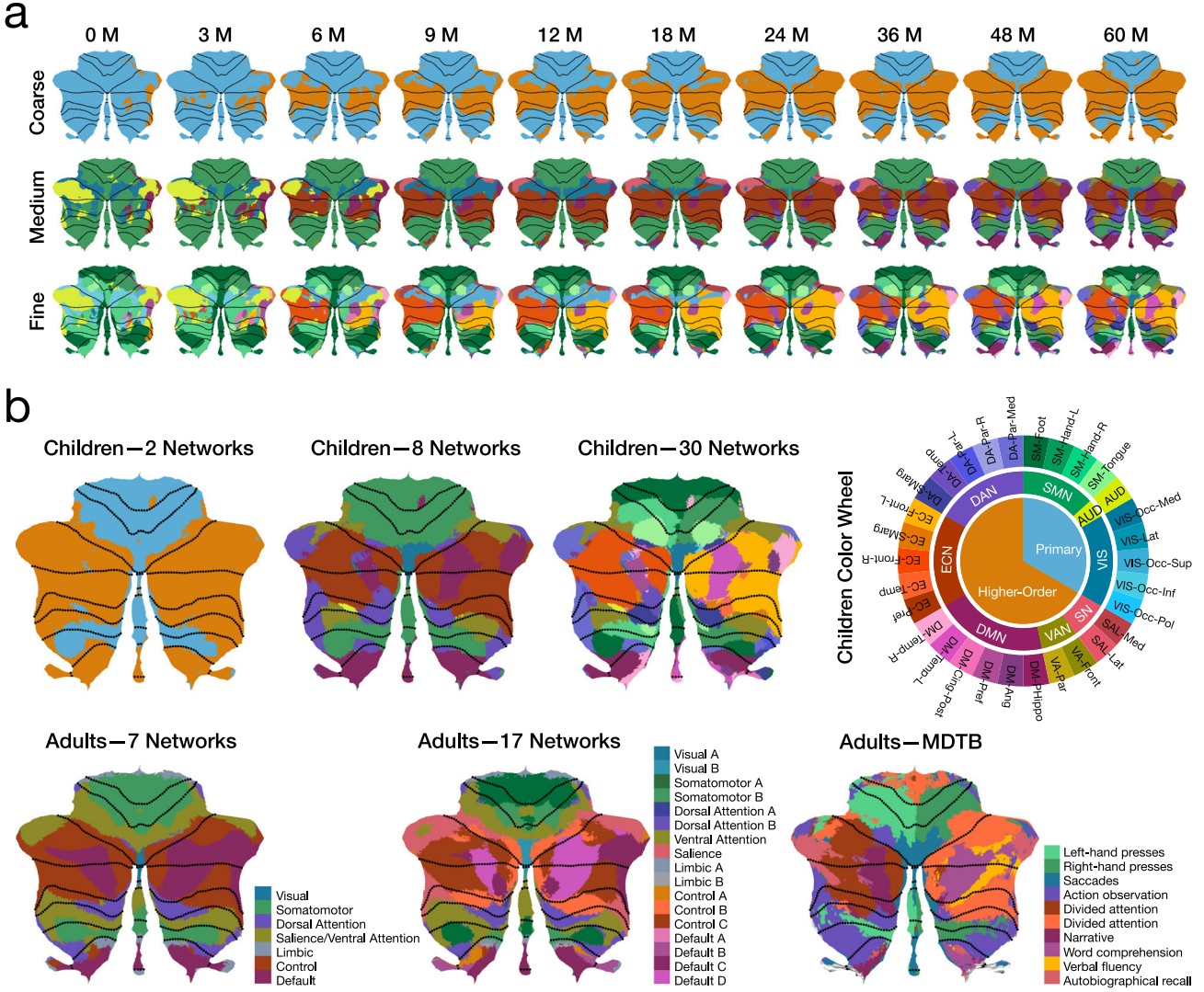

**Fig. 4 | Comparison of parcellation maps between children and adults.**
**a** Spatiotemporal patterns of cerebellar functional topography in early children at coarse, medium, and fine granularity. **b** Parcellation maps for children at 60 months, Buckner et al.'s 7 and 17 cerebellar resting-state networks[36,43], and the MDTB[44] parcellation map for adults.

involvement of the SMN, AUD, and VIS. Notably, they also exhibit increased engagement with the DAN but reduced involvement with the VAN and DMN. These findings suggest a developmental shift toward greater functional specialization as the cerebellocortical system matures, aligning with the evolving cognitive and sensorimotor demands of early childhood.

It is perhaps unsurprising that the cerebellar functional parcellation map of 60-month-old children resembles the parcellation map of adults based on resting-state functional networks (Fig. 4b, Adults–7 Networks and 17 Networks). Interestingly, the cerebellar functional parcellation map of 60-month-old children at fine granularity (Fig. 4b, Children–30 Networks) is similar to the adult MDTB parcellation map (Fig. 4b, Adults–MDTB). Specifically, the children's SM-Hand-R region largely overlaps with the "left-hand presses" region in the adult MDTB map, the SM-Hand-L region overlaps with the "right-hand presses" region, the VIS-Occ-Med region overlaps with the "Saccades" region, and the DA-Par-Med region overlaps with the "action observation" region. Additionally, the DM-Pref region corresponds with the "Narrative" region, and the DM-Temp-L region corresponds with the "word comprehension" region in the adult MDTB map. These findings suggest that early cerebellar functional organization in children is closely linked to specialized tasks,

indicating that the cerebellum may develop complex functional mappings earlier than previously thought.

Collectively, these findings demonstrate the dynamic evolution of cerebellar functional topography in early childhood, reflecting the increasing alignment with adult patterns as cognitive development and sensorimotor integration progress.

### Cerebellar functional gradients

Macroscale gradients of functional connectivity organize systematic information into abstract representations, providing valuable insights into how function varies across space[41,46]. Using gradient-based analysis, Guell and colleagues have established the spatial macroscale gradients of the cerebellum in adults[23,24,47]. However, despite the rapid neurodevelopment that occurs during childhood, the emergence and maturation of functional gradients in the cerebellum remain poorly understood. To address this gap, we employed LittleBrain[48] to generate visualizations based on cerebellar functional gradient maps, complementing the functional parcellation maps by capturing subtle and gradual spatial changes in cerebellar function. LittleBrain creates a two-dimensional map of cerebellar voxels, with each axis of the map representing a principal gradient: Gradient 1, which transitions from unimodal (motor) to transmodal (DMN) regions, and Gradient 2, which

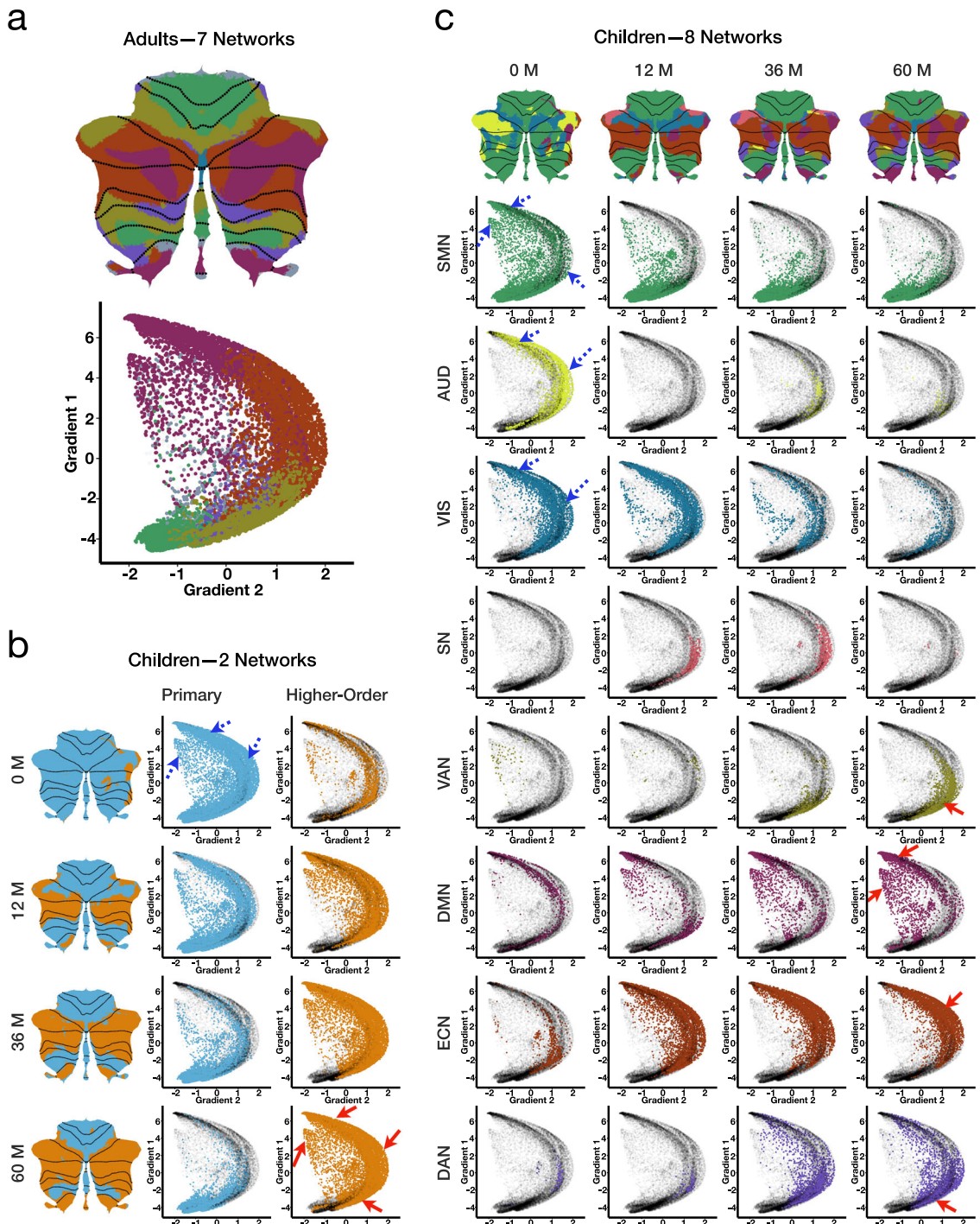

**Fig. 5 | Functional gradients. a** Cerebellar functional gradients in adults[36,43,48]. **b** Gradient maps of primary and higher-order networks at birth, 12 months, 36 months, and 60 months. **c** Gradient maps of large-scale networks at birth, 12 months, 36 months, and 60 months. Gradient maps were generated using LittleBrain[48], with each point representing a cerebellar voxel in the MNI space. Regions marked by blue dashed arrows on the gradient maps indicate areas where network points decrease or disappear with age, while regions marked by red solid arrows indicate areas where network points increase with age. Color schemes match Fig. 4b: Adults−7 Networks and Children Color Wheel. The cerebellar gradient maps of large-scale networks spanning the entire early childhood period are shown in Fig. S9.

shifts from task-unfocused (mind-wandering, goal-undirected thought) to task-focused (attentive, goal-directed thought) processing areas. Cerebellar voxels that are adjacent in the gradient map share similar functional connectivity patterns. We used the adult cerebellar functional parcellation from refs. 43,48 as the reference for our gradient-based analysis (Fig. 5a).

We investigated how functional topography evolves over time by mapping our cerebellar parcellation maps to gradient maps. At coarse granularity (Fig. 5b), primary networks in early childhood initially cover nearly the entire map but gradually concentrate in regions with low Gradient 1 and low Gradient 2 as age increases. In contrast, higher-order networks, initially confined to a small region in middle Gradient

2, progressively expand into areas with high Gradient 1 and high Gradient 2 over time.

At medium granularity (Fig. 5c and Fig. S9), the functional topography of different large-scale networks evolves differently with age:

- The SMN primarily varies along Gradient 1. With age, the presence of SMN in high Gradient 1 diminishes. By 60 months, the SMN is predominantly concentrated in low Gradient 1 and low Gradient 2, in alignment with the SMN gradient pattern observed in adults.
- Although AUD initially occupies a large portion of high Gradient 1 and high Gradient 2 at birth, it declines thereafter. After 6 months, AUD appears only sparsely on the map. Moreover, the points associated with the AUD and the VIS on the gradient maps reduce with age. Given the absence of AUD and the near-absence of VIS in the cerebellar gradient maps of adults, we speculate that the decline of AUD and VIS in children's cerebellar gradient maps reflects a developmental reorganization of cerebellar connectivity, with decreasing involvement in auditory and visual processing and increasing engagement in higher-order cognitive functions.
- The DMN and ECN are present on the gradient map from birth and vary along Gradients 1 and 2, respectively. As age progresses, the distribution of the DMN becomes concentrated in high Gradient 1, while the ECN distribution becomes concentrated in high Gradient 2, resembling the adult pattern.
- Both the DAN and VAN occupy only small areas on the gradient map at birth. However, beginning at 24 and 36 months, respectively, the DAN and VAN gradually shift to low Gradient 1 and slightly higher Gradient 2 regions, resembling their adult locations, and progressively expand over time.
- Since its emergence on the gradient map around 9 months, the SN is primarily concentrated in middle Gradient 1 and high Gradient 2, although its extent varies over time. Interestingly, this area intersects with the DMN, VAN, DAN, and ECN, underscoring the SN's pivotal role in network switching[49,50].

Together, these gradient maps illustrate that functional topography of large-scale networks evolves dynamically during early childhood. With development, most networks transition from a diffuse to a more focal distribution on the gradient maps, ultimately resembling adult-like patterns. Despite network-specific variations in cerebellar gradient maps during early childhood, peak concentration locations tend to stabilize around 36 months and align more closely with adult patterns. This stability highlights consistent functional organization across development, with core functional areas retaining fundamental patterns even as overall activity distributions become more refined.

## Temporal trends

We charted temporal changes in cerebellar functional topography based on the volume fraction of each network determined based on winner-take-all parcellation (Fig. 6a). At coarse granularity, although the proportion of cerebellar regions connected to primary networks is substantially larger than that of regions connected to higher-order networks at birth, it gradually decreases throughout early childhood. In contrast, the proportion of cerebellar regions connected to higher-order networks increases over time, gradually surpassing that of primary networks from approximately 24 months onward (Fig. 6a, left column).

At medium granularity, cerebellar regions connected to the SMN, AUD, and VIS decrease, while those connected to the DMN, DAN, and VAN generally increase throughout early childhood. Regions connected to the ECN demonstrate a distinct developmental trajectory: they expand rapidly and reach their volumetric peaks before 24 months—earlier than other higher-order networks—then remain relatively stable, with a slight decline thereafter. Notably, after 6 months of age, regions connected to the ECN are consistently larger than those of

any other network, except the SMN. We also observed that from birth to 6 months, most cerebellar regions are connected to the SMN, AUD, and VIS; from 6 to 24 months, the majority are connected to the SMN, ECN, and VIS; and from 24 to 60 months, the majority are connected to the SMN, ECN, and either DMN or DAN (Fig. 6a, right column). These developmental patterns suggest a dynamic shift in the cerebellum's role from primarily sensory-motor processing to a gradual and increasing integration with higher-order networks, such as the ECN, DMN, and DAN, indicating a diversification of cerebellar functions during early childhood.

Given that the winner-take-all approach assigns each voxel exclusively to a single network, it may fail to capture overlapping connectivity patterns that reflect functionally relevant interactions. To overcome this limitation, we also calculated the volume fraction of cerebellar voxels that show positive connectivity with each network (Fig. 6b). At coarse granularity, the volume fraction of primary networks generally declines from 0 to 12 months and remains relatively stable thereafter. In contrast, the volume fraction of higher-order networks shows a slight decline from 0 to 6 months, followed by a sustained increase until 48 months. At medium granularity, the volume fractions of the SMN and AUD exhibit an overall decline from 0 to 18 months before stabilizing, whereas that of the VIS continuously decreases throughout early childhood. The volume fractions of the SN and ECN increase during the first 24 months but decline thereafter. The volume fraction of the DMN exhibits a decrease during the first 12 months, followed by an increase, whereas the volume fractions of the VAN and DAN decline during the first 6 months and subsequently increase from 6 months onward. Notably, during the first 6 months of life, changes in volume fractions based on positive connectivity are relatively small for most networks, compared to those observed with the winner-take-all approach. This difference occurs because the winner-take-all method prioritizes the strongest connectivity, making it more sensitive to rapid local changes in network affiliations. This effect is especially pronounced during the first 6 months, a critical period of rapid cerebellocortical reorganization. Therefore, these two methods offer complementary perspectives on the development of cerebellocortical connectivity.

These results reveal dynamic changes in cerebellocortical functional connectivity during early childhood, showing how interactions between cerebellar regions and cortical networks evolve over time and highlighting the cerebellum's adaptive role in processing sensorimotor and cognitive information during this period.

## Functional lateralization

The functional lateralization of primary motor and higher cognitive functions in the cerebellum is well established, with cerebrocerebellar circuits considered to play a crucial role[51]. However, longitudinal research on cerebellocortical connectivity functional lateralization in developing children is currently lacking. To address this gap, we investigated cerebellar laterality with respect to the primary and higher-order networks from 0 to 60 months. We calculated the laterality index by counting the voxels in the left and right cerebellar hemispheres that were significantly connected to either higher-order or primary networks. We observed a consistent pattern of leftward lateralization in primary networks and rightward lateralization in higher-order networks, which stabilizes after 4 months of age (Fig. 6c).

## Sex differences

Sex differences in cerebellar gray and white matter volumes has been well-established in children and adolescents[52–54]. However, sex differences in cerebellocortical connectivity during early childhood are not well understood. To address this, we analyzed connectivity separately in female and male children to identify early sex-specific patterns. We found that cerebellar regions exhibiting strong connections with each RSN are generally consistent between female (Fig. S4) and male

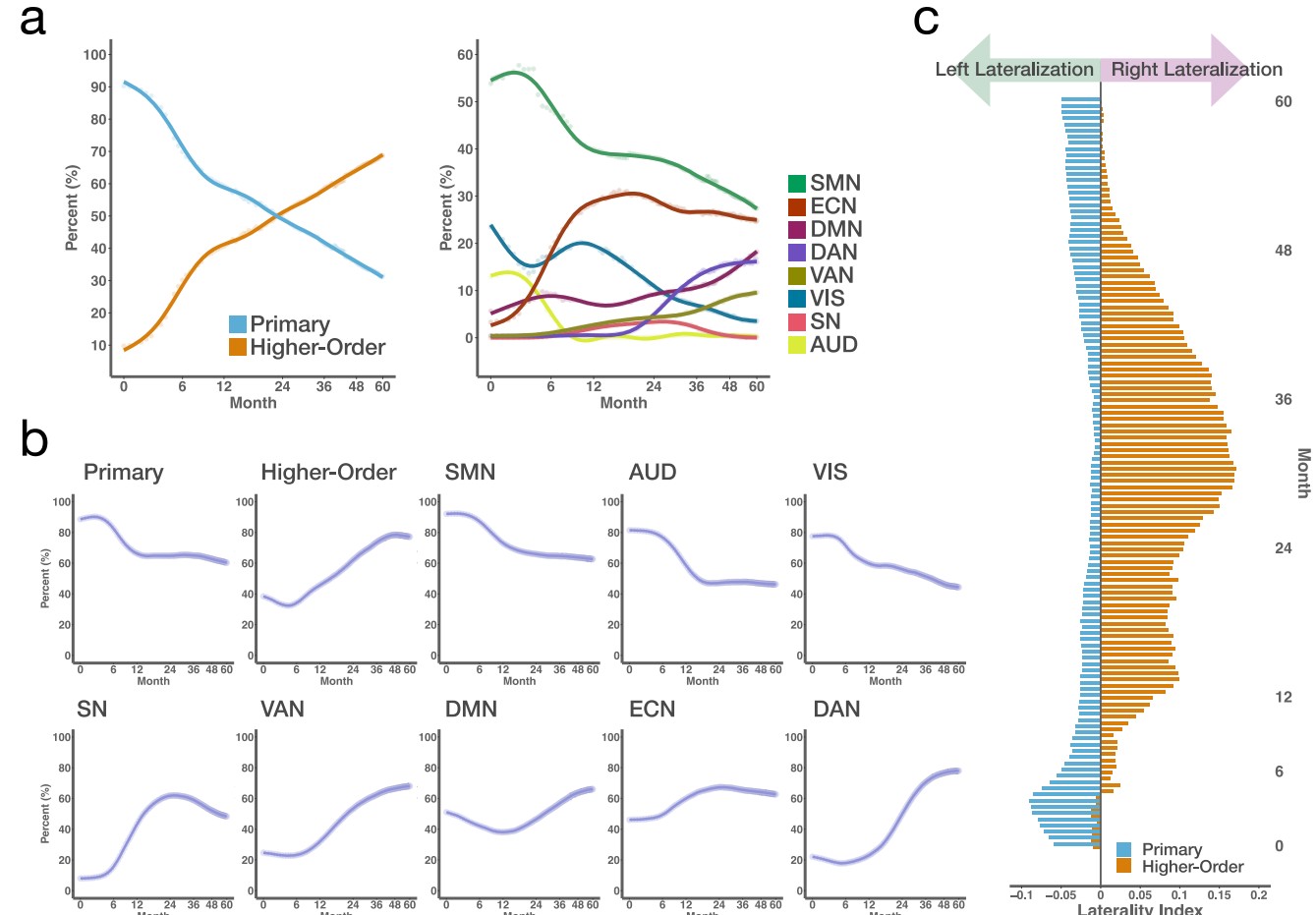

**Fig. 6 | Developmental patterns of cerebellar functional topography.**
**a** Trajectories of network-specific volume fractions at coarse granularity (left) and medium granularity (right). The curves show percentage change trends from generalized additive models; 95% confidence intervals are shaded (narrow and mostly not visible). **b** Early childhood trajectories of volume fractions of cerebellar voxels with positive connectivity to each cortical network. **c** Laterality of primary networks and higher-order networks from 0 to 60 months, with laterality index above 0 signifying right lateralization and below 0 signifying left lateralization. Primary, primary networks; Higher-order, higher-order networks; SMN sensorimotor network, AUD auditory network, VIS visual network, SN salience network, VAN ventral attention network, DMN default mode network, ECN executive control network, DAN dorsal attention network (Source data are provided as a Source Data file).

(Fig. S5) children. To better illustrate sex differences during early childhood, we generated spatial difference maps (Fig. S6), highlighting sex-specific variations in cerebellar functional connectivity for each RSN, and examined the temporal trajectories of peak cerebellar connectivity (Fig. S7) in both sexes.

We generated cerebellar functional parcellation maps for female and male children (Fig. 7a) to assess whether cerebellar functional topography exhibits sex-specific patterns. It can be observed from the coarse-granularity parcellation maps (Fig. 7a, top two rows) that the hierarchical organization of primary and higher-order networks emerged earlier in female children than in male children. The medium-granularity parcellation maps (Fig. 7a, bottom two rows) show that in females, Crus I and II are strongly connected to the ECN and DMN at birth, while in males, these connections become predominant around 9 months. We also generated cerebellar functional gradient maps for the eight large-scale networks, separately for females (Fig. S10) and males (Fig. S11), based on the parcellation maps. As expected, these gradient maps are generally consistent with age across sexes, though some networks show persistent sex-related differences.

We further mapped the temporal trajectories of network volume fractions at coarse and medium granularity for male and female children (Fig. 7b). At coarse granularity (Fig. 7b, left two columns), cerebellar regions connected to primary networks dominate at birth for both sexes, particularly in males. By around 24 months, cerebellar regions connected to higher-order networks become predominant. At medium granularity (Fig. 7b, right two columns), SMN-connected regions consistently dominate in both sexes. In females, ECN connectivity ranks second throughout, except in the first month. In males, ECN becomes second only after 12 months, with AUD or VIS ranking higher earlier on. Our findings highlight both shared and divergent patterns of cerebellocortical connectivity in male and female children during early childhood.

## Discussion

While extensive research has clarified the motor and cognitive functions of the cerebellum in adults[23,43,55,56], these roles remain poorly understood in early childhood—a critical period of rapid neural development. This stage involves ongoing synaptic pruning[18,57], which refines neural connections in response to development and stimuli, presenting a crucial window for neurodevelopmental intervention. Our study examined cerebellar connectivity with cortical networks and mapping its functional topography in children aged 0 to 60 months. Our findings show that cerebellar connectivity to higher-order networks is present at birth and generally strengthens throughout early childhood. The cerebellar representations of most cortical networks shift from broad, diffuse patterns in early childhood to more focused,

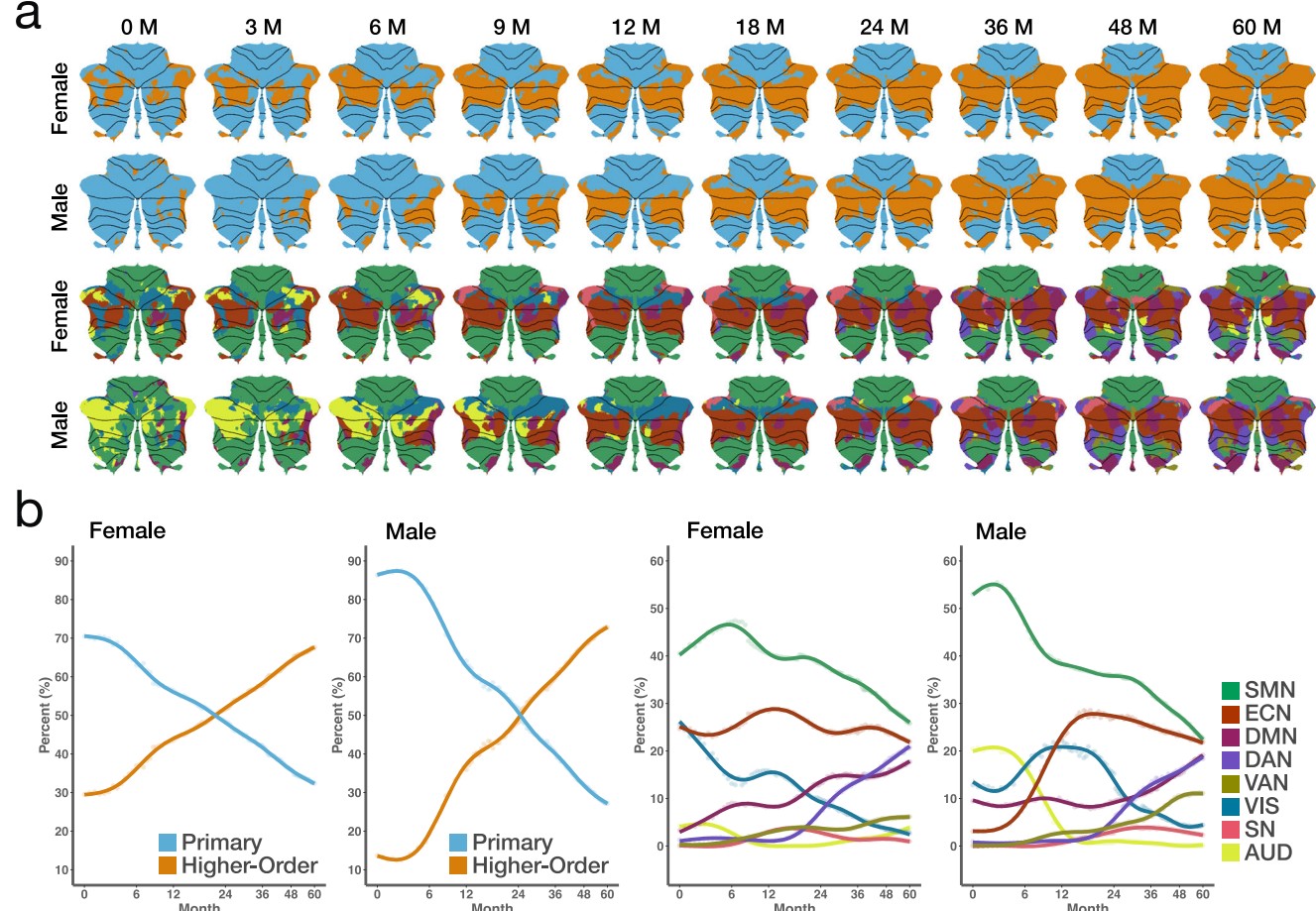

**Fig. 7 | Spatiotemporal patterns of functional topography for female and male children. a** Parcellation maps for female and male children at coarse granularity (top two rows) and medium granularity (bottom two rows). **b** Trajectories of network-specific volume fractions for female and male children at coarse granularity (two left plots) and medium granularity (two right plots). The curves show percentage change trends from generalized additive models; 95% confidence intervals are shaded (narrow and mostly not visible). (Source data are provided as a Source Data file).

adult-like spatial organization. We explored the hierarchical organization of cerebellocortical connectivity in early childhood and found that by around 36 months, cerebellar topography begins to resemble the adult pattern, becoming increasingly similar over time. We observed lateralized patterns of cerebellar connectivity, with higher-order networks being right-lateralized and primary networks being left-lateralized, further highlighting the cerebellum's functional asymmetry. We identified male and female patterns in both the organization and development of cerebellocortical functional connectivity. These findings illustrate how cerebellocortical interactions develop over the first 5 years of life, enhancing our understanding of the cerebellum's functional roles, topographical organization, asymmetries, and sex-specific developmental patterns, and highlighting the need for further research.

The cerebellum plays a well-documented role in motor control and sensory modulation in both humans[58] and other species[59], coordinating complex actions such as skilled finger movements, eye positioning, and control of the trunk and limbs. It is also crucial for motor learning[60], essential for acquiring and refining motor skills. Damage to the cerebellum's auditory and visual regions reduces accuracy in auditory- and visual-evoked orienting behaviors[59], highlighting the cerebellum's role in sensory processing. Cerebellar injury can impair visual target tracking[61] and motion detection[62], underscoring the cerebellum's importance in sensory-motor integration. Despite these findings, few studies have examined functional connectivity between the cerebellum and motor cortex in young children[27,63], and

neuroimaging evidence of cerebellar connections with the auditory (AUD) and visual (VIS) networks remains scarce. Our study confirms that the developing cerebellum functionally connects with both the SMN and sensory regions, such as the AUD and VIS. However, due to the lack of ground truth validation from optogenetics, histology, or tractography, these findings in early childhood remain to be fully confirmed. Notably, the auditory cortex's close proximity to the sensorimotor cortex has often caused its distinct cerebellar connections to be overlooked. Recent research[64] identified unique cerebellar functional connectivity with the auditory cortex, separate from its connectivity with the SMN. Our study adds to this understanding by showing that cerebellar functional connectivity with the AUD differs from that with the SMN not only in strength but also in developmental patterns, underscoring the cerebellum's distinct role in auditory processing during early development. Moreover, we found strong cerebellar functional connectivity with the VIS in infants, in sharp contrast to the much weaker connectivity seen in adults, which is largely confined to the cerebellar vermis[26,38]. This difference may reflect the early dominance of visual, auditory, and sensorimotor cortical hubs in childhood[65].

While there is general agreement that the cerebellum is functionally connected to primary networks, especially motor networks, its connectivity with higher-order networks during early childhood, particularly in infancy, remains uncertain. Research in this area is limited. While one study using diffusion tensor tractography suggests that structural connectivity between the cerebellum and

higher-order cortical regions is established during infancy[66], another functional MRI study found no evidence of cerebellar functional connectivity with these networks during this period[27]. This discrepancy highlights the complexity of cerebellocortical interactions in early childhood and underscores the necessity for further investigation. A recent study[67] provides critical insight into these interactions, revealing that neonatal cerebellar injury can disrupt the development of higher-order cortical regions. Beyond motor impairments, neonates with isolated cerebellar injuries may also exhibit delays in cognitive, language, and social abilities[68,69], suggesting crucial functional interactions between the cerebellum and higher cognitive networks during early life.

Infant learning was perceived as a passive accumulation of sensory experiences, predominantly associated with the sensory and motor cortex. However, emerging research[70] suggests that infants are intrinsically motivated for active learning, with the prefrontal cortex playing a key role in this process. Our study corroborates this view by revealing widespread cerebellar connectivity with higher-order networks during early childhood. Although connectivity with higher-order networks is initially weaker than with primary networks, it generally strengthens with age. Notably, specific higher-order networks, especially the ECN located in the frontal cortex, exhibit robust connectivity with the cerebellum from birth. This finding is consistent with our previous research[71], which highlighted strong structural connections between the cerebellum and the superior frontal gyrus—one of the key regions of the ECN. Our findings confirm that the functional connectivity between the cerebellum and higher-order networks is established early in life and may play a crucial role in shaping cognitive and behavioral functions during childhood, potentially influencing the developmental trajectories of these networks into adulthood. This early interaction may shed light on the underlying mechanisms of early-onset cognitive disorders such as ASD and ADHD, which are linked to abnormal cerebellar connections with higher-order networks[29–34], offering insights into how cerebellar development affects later cognitive functions.

The contralateral organization between the cerebellum and both the motor and prefrontal cortices is well-established in adults[19,20]. Our findings show that this organization is already present in early childhood, supporting the existence of the cross-hemispheric cerebrocerebellar loop[72]. Identifying this loop in early childhood may help explain contralateral developmental abnormalities in the cerebellum after unilateral cerebral injury in infancy, potentially caused by remote trans-synaptic effects[8]. In addition to contralaterality, we observed a hierarchical organization of primary and higher-order networks within the cerebellum, closely following the S-A axis[41]. Extending from primary sensorimotor and visual regions to higher-order association areas[73,74], the S-A axis reflects a fundamental principle of hierarchical cortical organization, with spatiotemporal pattern of cortical maturation proceeding hierarchically along this axis[73,75]. While the hierarchical organization along the S-A axis is well established in the cerebral cortex, there is limited research on whether a similar organization exists in the cerebellum. Guell et al.[24] identified functional gradients in the cerebellum that mirror the cortical hierarchy, revealing a dual motor and triple nonmotor organization. However, unlike the S-A axis of cerebellocortical functional connectivity in adults[24,76]—where gradient boundaries are typically aligned with the superior posterior, prepyramidal, and secondary fissures—the first gradient boundary in children initially appears at the horizontal fissure a few months after birth. This boundary shifts gradually, reaching the superior posterior fissure by around 24 months, while the second and third gradient boundaries resemble those in adults at the prepyramidal and secondary fissures from around 9 and 36 months of age, respectively. These findings suggest that the cerebellum's hierarchical organization of primary and higher-order functions begins to emerge in infancy and gradually aligns with the adult pattern along the S-A axis, possibly

reflecting the cerebellum's early and dynamic integration into broader cortical networks.

Distinct cerebellar gradient patterns have been observed in adults[24,25,48]: the SMN, ECN, and DMN occupy low, middle, and high positions on Gradient 1, respectively. The VAN and DAN are positioned along Gradient 1 away from the DMN, with the ECN situated between the VAN/DAN and DMN, acting as a mediator. Gradient 2 is thought to be related to task load, with low Gradient 2 corresponding to low task load networks, such as the motor network and DMN, and high Gradient 2 to task-positive networks, including the VAN, DAN, and ECN. At birth, the patterns are more diffuse, especially for higher-order networks, but gradually become more focal and adult-like with age. Since similar functions occupy adjacent gradient positions[77], the progressive focalization of these gradients reflects functional specialization. Notably, despite dynamic changes in cerebellar gradients during early childhood, the anchor positions of most networks become relatively stable from around 36 months, mirroring those in adults. This suggests the foundational framework is likely laid down at birth and refined with age, potentially through synaptic pruning or activity-dependent synapse refinement.

Studies on cerebellar functional asymmetry suggest that cognitive functions typically exhibit right laterality, while motor functions tend to show left laterality[51]. Consistent with this, our findings indicate that the cerebellum demonstrates rightward lateralization in connection with higher-order networks and leftward lateralization with primary networks throughout early childhood. Clinical studies[78,79] further support this pattern, revealing that damage to the right cerebellar hemisphere can lead to cognitive impairments, including deficits in language and literacy, whereas damage to the left cerebellar hemisphere is frequently associated with impairments in visuospatial and spatial skills. These findings highlight the role of lateralization in early development, revealing distinct left and right cerebellocortical connectivity patterns that may underlie their differing roles in cognition and motor control.

Although no studies have yet reported sex differences in cerebellocortical functional connectivity during development, such differences have been well documented in cerebellar volumetric studies involving infants, children, adolescents, and adults[52–54]. Our findings suggest that hierarchical organization in the cerebellum develops later in males than in females. Before the age of two, the proportion of cerebellar voxels predominantly connected to higher-order networks is greater in females than in males. This may contribute to a deeper understanding of the increased susceptibility of males to early-onset neuropsychiatric disorders[80], such as ASD[31] and ADHD[81], which have been linked to reduced cerebellar connectivity with higher-order networks. In addition, our study indicates that females exhibit noticeably stronger cerebellar connectivity with SM-Tongue than males during early childhood. This finding provides neuroimaging evidence supporting earlier speech production[82] and better verbal performance[83] in females. Our study underscores the importance of examining sex differences in early cerebellocortical functional connectivity, as they may impact cognitive and behavioral development. Additionally, it is crucial to recognize that factors such as genetics, nutrition, and socioeconomic conditions may also play a role in shaping functional connectivity[84]. Future research should account for these influences to better understand their contributions to cerebellocortical development and how they may interact with sex differences.

In summary, we investigated cerebellocortical functional connectivity during early childhood, spanning from birth to 60 months. Our findings demonstrate the presence of cerebellar connectivity not only with primary networks but also with higher-order networks, even at birth. Our parcellations with fine temporal resolution captured functional topography at different developmental stages, revealed the hierarchical organization of primary and higher-order networks, and suggested an S-A axis of early cerebellar functional development.

Furthermore, we uncovered lateralization and sex-specific patterns in early cerebellar functional organization. These findings may guide future research into the neural mechanisms underlying early brain development.

## Methods

### Participants

The data utilized in this study were collected as part of the UNC/UMN Baby Connectome Project (BCP)[35]. After data preprocessing and quality control, the final dataset included 1017 scans from 275 healthy participants (130 males and 145 females) ranging from birth to 60 months, with up to six longitudinal scans per participant. Parents received detailed information about the study's objectives before providing written consent. Ethical approval for all study procedures was granted by the institutional review boards of the University of North Carolina at Chapel Hill (UNC) and the University of Minnesota (UMN). Participants in the BCP study were financially compensated[35].

### Data acquisition

Children under 3 years old (0–36 months) were scanned while naturally asleep, without the use of sedatives. Prior to imaging, all babies were fed, swaddled, and fitted with ear protection. Children older than 3 years (36–60 months) were scanned while either asleep or watching a passive movie[35,85]. All images were acquired using 3T Siemens Prisma MRI scanners (Siemens Healthineers, Erlangen, Germany) with 32-channel coils at the Biomedical Research Imaging Center (BRIC) at UNC and the Center for Magnetic Resonance Research (CMRR) at UMN[35]. MRI acquisition parameters are summarized as follows:

- T1-weighted MR images were obtained with a 3D magnetization prepared rapid gradient echo (MPRAGE) sequence: isotropic resolution = 0.8 mm, field of view (FOV) = 256 mm × 256 mm, matrix = 320 × 320, echo time (TE) = 2.24 ms, repetition time (TR) = 2400/1060 ms, flip angle = 8°, and acquisition time = 6 min 38 s.
- T2-weighted MR images were obtained with a turbo spin echo (TSE) sequence: isotropic resolution = 0.8 mm, FOV = 256 mm × 256 mm, matrix = 320 × 320, TE = 564 ms, TR = 3200 ms, and acquisition time = 5 min 57 s.
- Resting-state functional MRI data were collected using a single-shot echo-planar imaging (EPI) sequence: isotropic resolution = 2 mm, FOV = 208 mm × 208 mm, matrix = 104 × 104, TE = 37 ms, TR = 800 ms, flip angle = 52°, and acquisition time = 5 min 47 s.

### Data preprocessing

The processing pipeline for structural and functional MRI data is summarized in Fig. S1. Structural MRI data were visually assessed by a board-certified neuroradiologist[35], and high-quality T1-weighted (T1w) and T2-weighted (T2w) images were selected for further processing. Structural MRI processing included N3 normalization[86] to remove intensity inhomogeneity, linear alignment of the T2w image to the T1w image using FLIRT[87], and deep-learning-based tissue segmentation using T1w and/or T2w images. The resulting tissue segmentation map facilitated the registration of rs-fMRI images to the structural MRI space. The fMRI blood oxygen level dependent (BOLD) data underwent minimal preprocessing as follows: (i) Head motion correction using the FSL `mcflirt` function[88], rigidly registering each fMRI time frame to a single-band reference (SBref) image and generating the corresponding motion parameter files; (ii) EPI distortion correction using FSL `topup` function[87,89], generating a distortion correction deformation field using a pair of reversed phase-encoded field maps; (iii) Rigid registration (6 degrees of freedom) of the SBref image to the field maps; (iv) Rigid boundary-based registration (BBR)[90] of distortion-corrected SBref image to the corresponding tissue-segmented structural image, using mutual information as the cost

function and initialized with a prealignment; and (v) One step sampling using combined deformation fields and translation matrices, producing motion- and distortion-corrected fMRI data in the subject's native (structural MRI) space.[91]

### Data denoising

The minimally preprocessed fMRI data were further denoised prior to subsequent analyses. Slow signal drift was removed by detrending the data using a high-pass filter with a cutoff frequency of 0.001 Hz. Residual motion artifacts were mitigated using ICA-based automatic removal of motion artifacts (ICA-AROMA)[92,93]. Specifically, we applied a 150-component independent component analysis (ICA) and classified each component as either BOLD signal or artifact based on high-frequency contents, correlation with realignment parameters from head motion correction, edge effects, and CSF fractions. Motion-related artifacts are then non-aggressively regressed out. The denoised fMRI data were subsequently mapped to the Montreal Neurological Institute (MNI) space[94,95]. We then applied Gaussian smoothing with a full-width at half-maximum (FWHM) of 4 mm to the cerebrum and cerebellum separately. The cerebral and cerebellar masks were obtained by segmenting the high-resolution MNI-ICBM 152 symmetrical template[94–96] and then downsampling to 2 mm resolution. Finally, we intensity-normalized the data to a constant mean volume intensity of 10,000[97], reducing scanner-related intensity variations and enabling reliable cross-subject comparisons for subsequent functional connectivity analyses.

### Quality control

To ensure good data quality, we computed framewise displacement (FD) from the motion parameters obtained via head motion correction. Samples with mean Power's FD (absolute sum of motion parameters) exceeding 0.5 mm—a common threshold used in fMRI studies[98]—were excluded. Additionally, we set a limit for Jenkinson's FD[88] (Euclidean sum of motion parameters) at 0.2 mm to further minimize the impact of motion-affected samples. Out of 1656 preprocessed fMRI samples, 1493 met Power's FD criterion, and 1278 further satisfied Jenkinson's FD criterion. We then visually evaluated the quality of skull stripping, tissue segmentation, and fMRI-to-T1w image registration, retaining 1245 with satisfactory quality. The fMRI data in MNI space were categorized based on geometric distortions as "pass" (no or light distortion), "questionable" (light to medium distortion), and "fail" (medium to large distortion). After visual inspection, 1017 samples were labeled as having "questionable" or better quality, of which 703 were marked as "pass". Samples labeled as "fail" were excluded from all analyses. Only "pass" samples were used to compute template spatial maps, minimizing the influence of artifacts on functional network estimation. All 1017 samples with at least "questionable" quality were included in subsequent analyses.

### Independent component analysis and network identification

We performed probabilistic group independent component analysis (GICA) exclusively on the "pass" fMRI data to minimize noise-related components. This yielded 40 resting-state networks (RSNs), which were used as templates for subsequent analyses. An anatomical review of the template RSNs showed that 30 were associated with the cortex, while ten were linked to the cerebellum or subcortical regions. We further categorized the 30 cortical RSNs into eight large-scale networks[20,99] (Table S1): SMN, AUD, VIS, SN, VAN, DMN, ECN, and DAN. RSNs within the SMN were named according to major motor functions, while those in other large-scale networks were named based on their anatomical locations. The terms "left" or "right" designate the hemisphere in which each component is situated.

### Individual RSNs via dual regression

Using the template RSNs, we computed the RSNs for each individual based on their denoised fMRI data via dual regression[100]. We then

applied Gaussian mixture modeling (GMM) to each individual RSN to generate activation probability maps. These maps were used to create subject-specific binary masks of cerebral RSNs, which were then used to compute cerebellocortical functional connectivity in subsequent analyses.

## Cerebellocortical functional connectivity

Cerebellocortical functional connectivity (FC) was computed for each voxel in the cerebellum with respect to a set of cortical functional networks. A Butterworth bandpass filter (0.008–0.1 Hz) was applied to the BOLD time series[101,102]. For each of the 30 cortical RSNs, we extracted a representative time series by computing the primary eigen time series of the denoised BOLD signals from the top 10% most activated voxels within the cerebral mask of the RSN[19]. This eigen time series corresponds to the first eigenvector obtained via singular value decomposition (SVD), with its sign adjusted[103] to match the mean voxel time series. To prevent signal leakage between the cerebrum and cerebellum, cerebral voxels overlapping with the cerebellar mask—dilated by $8 \times 8 \times 8\ mm^3$—were excluded from the calculation. FC was quantified as partial correlation scores between the cortical eigen time series and the denoised BOLD time series of each cerebellar voxel within the non-dilated cerebellar mask. The correlation scores were transformed using Fisher's r-to-z transformation and then standardized to unit variance.

## Spatiotemporal maps

We used a generalized additive mixed model (GAMM), implemented with the `gamm4` package in R[104], to model cerebellocortical FC at each voxel. The model included a smooth age term and random effects for subject ID and scan site: $Y \sim s(age, k = 10) + (1|SubjectID) + (1|SiteID)$, where $s(\cdot)$ represents the smooth term and $(1|\cdot)$ models random effects. The number of basis functions, $k$, was empirically set to slightly exceed the degrees of freedom of the data. Three separate GAMMs were fitted, accounting for females, males, and both sexes. We generated cerebellocortical FC spatial maps from birth to 60 months of age at half-month intervals. For each predicted FC map, we applied Gaussian mixture modeling using FSL MELODIC to estimate the null distribution parameters—specifically, the mean, variance, and proportion of null voxels. We then aggregated the estimated null parameters across all time points to compute a global mean and variance of the null distribution for each cerebellocortical FC component. These values were used to z-normalize the corresponding FC maps, allowing for meaningful comparisons across cerebellocortical FC components and developmental time points. Peak FC was defined as the 99th percentile value of the FC spatial map. Medium- and coarse-granularity spatial maps were obtained by taking the signed absolute extrema of the fine-granularity maps.

To map sex differences, we subtracted the male FC maps from the female FC maps and standardized the differences by dividing by $\sqrt{2}$, yielding maps with unit null variance. This results in standardized z-score maps, where a value of 3.1 corresponds approximately to a one-tailed $p$ value of 0.001.

## Parcellation

Parcellation maps were created by applying a winner-take-all approach to the cerebellocortical FC spatial maps of the 30 RSNs. First, we applied 7 mm FWHM spatial smoothing to each FC map to further reduce noise. Next, we used probabilistic threshold-free cluster enhancement (pTFCE)[105] to generate enhanced $p$ value maps, which were upsampled to a resolution of $0.5 \times 0.5 \times 0.5\ mm^3$. Using the winner-take-all approach, we assigned each cerebellar voxel to the FC component with the lowest enhanced $p$ value. The resulting fine-granularity parcellation maps were then grouped according to Table S1 to create medium- and coarse-granularity maps. Developmental changes in parcel volume fraction, relative to the total cerebellar volume, were modeled using thin-plate splines via the `gam` function in

R: `volume_fraction ~ s(age)`, with $age = \log_2\left(\frac{month}{12} + 1\right)$. A logarithmic time scale was used to capture rapid changes during the first year of life.

## Functional laterality

Functional laterality was quantified based on the volume fractions of voxels in the right and left cerebellar hemispheres exhibiting FC values greater than 3. The laterality index (LI) was calculated using the following formula: $LI = \frac{R-L}{R+L}$, where $R$ and $L$ represent the volume fractions in the right and left cerebellar hemispheres, respectively. The LI ranges from −1 (total leftward lateralization) to 1 (total rightward lateralization), with 0 indicating complete symmetry.

## Effects of wakefulness and motion

We evaluated whether the unexplained variance in the GAMM residuals could be attributed to wakefulness in children over 3 years old and motion artifacts. To this end, we fit the following linear model at each cerebellar voxel: `residuals ~ wakefulness + motion`, where `residuals` are the residuals from the GAMM after accounting for age and random effects; `wakefulness` is a binary variable indicating whether the subject is older than 3 years; and `motion` is a continuous variable representing the mean framewise displacement of the fMRI scan[98]. For each model, two-tailed t-tests were used to assess the significance of the regression coefficients ($\beta$) associated with `wakefulness` and `motion`. The test statistic was calculated as $\beta/SE(\beta)$, where SE denotes the standard error of the estimate. The percentage of voxels exhibiting significant effects ($p < 0.01$) for `wakefulness` and `motion` are reported in Table S2.

## Functional correlation with Mullen scores

We investigated the relationship between cerebellocortical FC and scores from the Mullen scales of early learning (MSEL)[106] (Table S3), using the following steps: (i) Normalize individual cerebellocortical FC spatial maps to have a null distribution of zero mean and unit variance. (ii) Spatially smooth (7 mm FWHM) spatial maps to reduce noise. (iii) Consolidate medium- and coarse-granularity spatial maps using signed absolute extrema. (iv) Extract the peak FC of each spatial map. (v) Adjust peak FC and Mullen scores for age effects with a linear mixed model (LMM): $Y \sim age + (1|subject\ ID)$. (vi) Compute Pearson correlations between LMM residuals of peak FC and Mullen scores.

## Visualization

Cortical functional networks were visualized using MRIcroGL (https://www.nitrc.org/projects/mricrogl/). Cerebellocortical FC flat maps were displayed using the Statistical Parametric Mapping software package (SPM12, https://www.fil.ion.ucl.ac.uk/spm/software/spm12/) and the SUIT[36] software package (https://github.com/jdiedrichsen/suit). Functional gradient maps were generated with LittleBrain[48] (https://github.com/xaviergp/littlebrain). Trajectories of cerebellocortical FC across age were analyzed in R and visualized using `ggplot2`[107].

## Reporting summary

Further information on research design is available in the Nature Portfolio Reporting Summary linked to this article.

# Data availability

The BCP dataset used in this study is available from the National Institute of Mental Health Data Archive (NDA, https://nda.nih.gov). Processed data supporting the findings of this study is available at Figshare (https://doi.org/10.6084/m9.figshare.25970518). Source data are provided with this paper.

# Code availability

Software tools used in this work include FSL 6.0, ICA-AROMA, Advanced Normalization Tools (ANTs 2.2.0, https://github.com/

ANTsX/ANTs), MATLAB 2023a (toolboxes include SPM12 https://www.fil.ion.ucl.ac.uk/spm/software/spm12/, and SUIT https://github.com/jdiedrichsen/suit), LittleBrain (https://github.com/xaviergp/littlebrain), Python 3.12, R (4.4.0, primarily gamm4, mgcv, bigmemory, ggplot2, fslr, pTFCE, oro.nifti, R.matlab, matlib, dplyr and pryr), and MRIcroGL (1.2, https://www.nitrc.org/projects/mricrogl/).

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

## Acknowledgements

The authors thank Jörn Diedrichsen, Caroline Nettekoven, and Aikaterina Maroli for their valuable discussions. This work was supported in part by the United States National Institutes of Health (NIH) under grants R01 MH125479, R01 EB008374, R01 EB035160, and R01 MH133836 (P.-T.Y.).

## Author contributions

W. Lyu: Methodology, investigation, visualization, data curation, writing—original draft, and writing—review and editing. K.-H.T.: Data curation, methodology, software, visualization, and writing—review and editing. K.M.H.: Resources and writing—review and editing. L.W.: Resources. W. Lin: Resources. S.A.: Resources and writing—review and editing. P.-T.Y.: Conceptualization, supervision, funding acquisition, investigation, validation, and writing—review and editing.

## Competing interests

The authors declare no competing interests.
