## [Transparent Peer Review file · Nature Communications]

Functional Development of the Human Cerebellum from Birth to Age Five

Corresponding Author: Professor Pew-Thian Yap

Version 0:

Reviewer comments:

Reviewer #1

(Remarks to the Author)

The study explores the functional development of the cerebellum and its connectivity with the cerebral cortex from birth to 60 months in children, utilizing over 1,000 resting-state fMRI scans. Findings reveal that the cerebellum is involved in higher-order cognitive functions early in life, highlighting a hierarchical organization and functional asymmetry between the left and right hemispheres. Notably, the research identifies sexual dimorphism in cerebellar development, indicating differing connectivity patterns between male and female children.

This article provides a thorough analysis of the development of cerebellar-cortical connectivity, clearly describing the relevant findings during the neonatal phase. The structure is logical and well-organized, offering deep insights into the cerebellum's crucial role in cognitive processes, especially beyond traditional motor functions. These findings not only offer valuable insights into early childhood development but also lay the groundwork for future research in this area. However, there are still some minor questions that need further exploration to enhance the understanding of the interactions between the cerebellum and the cortex.

1. While the introduction emphasizes the cerebellum's role in higher-order cognitive functions and its potential connection to neurodevelopmental disorders like ASD and ADHD, there is a noticeable gap between these claims and the actual analysis, which does not include direct cognitive or disease-specific data. Additionally, the rationale for investigating cerebellar functional gradients is not fully elaborated. Providing more context and justification for why these gradients are important for early childhood development would strengthen the overall narrative.

2. The study makes important claims about the cerebellum's influence on cognitive and motor outcomes, yet the results section lacks a direct examination of these outcomes. Given that the BCP dataset includes individual behavioral and cognitive measurements, the authors could strengthen their conclusions by utilizing these data. Specifically, they could test whether the cerebellar-cortical connectivity (e.g., with networks like VIS or DMN) is associated with actual cognitive and behavioral performance. Additionally, exploring whether these associations vary across different behavioral domains would provide more robust evidence to support the claim of cerebellar connectivity's potential influence on cognitive and motor outcomes.

3. The use of a "winner-take-all" approach in the analysis may have the unintended effect of exaggerating the differences between groups. By assigning each voxel exclusively to the network with the strongest connection, this method might overlook subtler, yet meaningful, overlapping connectivity patterns and underrepresent variability within groups. This can lead to an overemphasis on the distinctions between developmental patterns (e.g. Fig 6a. volume fractions plot), potentially overstating the magnitude of sex differences (Fig. 7b).

4. In the analysis of sex differences, the authors present intriguing findings regarding cerebellar connectivity, but it would be beneficial to address potential confounding variables that may influence these results. Factors such as socioeconomic status, overall head motion, and other biological variables could also impact cerebellar development and connectivity. Without controlling for these variables, the observed differences between males and females might be influenced by factors beyond sex alone. These factors should be considered or the potential impact mentioned in the discussion.

5. The analysis of sex differences in cerebellar connectivity lacks sufficient statistical description. While differences between males and females are reported, there is limited information regarding the statistical tests used, effect sizes, or confidence

intervals that would help quantify these differences.

Reviewer #2

(Remarks to the Author)

This paper provides a valuable contribution to the small but growing literature on the developing cerebellum. Given that clinical findings have shown that prenatal and early-life damage to the cerebellum can cause devastating consequences for cognitive, social, and motor functions, investigations into the developing cerebellum are greatly needed. The authors have provided a valuable roadmap for the community. The quality of the data and analyses are very high. The conclusions they draw mostly stick to findings.

Critique

1. In the general discussion, instead of using the term “connectivity” please use “resting state connectivity” or “functional connectivity” more to remind readers that they are not looking at a true, structural connectivity. As an example, you state “While there is consensus on the existence of connectivity between the cerebellum and primary networks, uncertainty remains regarding its connectivity with higher-order networks.” This can be easily misconstrued; I suggest you write something like ““While there is consensus on the existence of resting state connectivity between the cerebellum and primary networks (at least motor networks), uncertainty remains regarding its resting state connectivity with higher-order networks.”
2. Related to my last point, you discuss visual and auditory functional connectivity. What is the ground truth for this? To my knowledge, there is nothing in optogenetics, histology, or tractography which to ground these findings. Please provide evidence or note the lack of ground truth evidence.
3. The fuzziness of your terminology is found in later parts of the General Discussion, where you discuss connectivity, but in this instance, you’re discussing gold standard, monkey tract-tracing. ““While there is consensus on the existence of connectivity between the cerebellum and primary networks, uncertainty remains regarding its connectivity with higher-order networks.” This fuzziness is acceptable in the resting state community; it is not acceptable in neuroscience literatures more connected to ground truth structural connectivity.
4. There is an incorrect statement in the General Discussion “Given that language processing is predominantly lateralized to the left cortical regions and visual functions are right-lateralized...” Visual functions are NOT right lateralized. Completely wrong so should be removed.
5. This statement lacks nuance to the point of being incorrect and your citations are way too old. “Sexual dimorphism in human brain structure and function is well documented.” Please find a few citations from the last 10 years. No, it’s not well documented in humans. There are subtle differences in some measures, but not others. Again, referring to ground-truth evidence (e.g. histology!) is the best evidence. If you don’t have that evidence, clearly lay out where the evidence is from.

Reviewer #3

(Remarks to the Author)

Our understanding of the typical connections between the cerebellum and the cerebral cortex, especially during the earliest phases of human development, is still limited. The authors utilized functional magnetic resonance imaging (fMRI) data from over 1,000 scans of 270 children, aged from birth to 60 months, as part of the Baby Connectome Project (BCP), to chart the connectivity patterns between the cerebellum and resting-state functional networks in the brain. The study aimed to track the evolution and structure of cerebellar-cerebral connectivity, with a particular focus on when and how higher-order network connections are established. The authors employed independent component analysis (ICA) to parcellate the cerebellum functions and also utilized the most recent gradient analysis techniques to characterize the functional S-A axis. Their findings indicated that the cerebellum is connected not just to primary networks but also to higher-order networks right from birth. The detailed parcellations with high temporal resolution allowed for the capture of functional topography at various stages of development, uncovering the hierarchical arrangement of primary and higher-order networks. They also proposed an S-A axis as a key developmental trajectory for early cerebellar function. Overall, these discoveries provide new insights into the early stages of cerebellocortical connectivity development. However, some major issues should be appropriately addressed.

I noticed that imaging was carried out on children under the age of 3 (ranging from birth to 35 months) during their natural sleep, without the need for sedation. For children above the age of 3 (from 36 to 60 months), the imaging procedures were conducted either while they were sleeping or while they were passively viewing a movie. That said the brain states for children below and above the age of 3 during scanning are totally different. They are not the same condition. In other words, natural sleep and passively viewing a movie are totally different and not the “true” resting state. They can not tell us the normal function of our cerebellum. So, I’m afraid we can not take them together to characterize normal developmental trajectory for early cerebellar function.

I’m not convinced by the present fMRI preprocessing, especially for cerebellum. The cerebellum should be separately preprocessed to guarantee the precise normalization, which is crucial to precisely parcellate the cerebellum function units and characterize the precise development trajectory of cerebellocortical connectivity in early stage. Traditional preprocessing steps are not suitable for precise cerebellum mapping. For example, regarding the Gaussian smoothing, a full-width half-maximum (FWHM) of 4mm was applied. Smoothing data for cerebral cortex and cerebellum should be separately conducted with cerebral mask or with cerebellum mask. Obviously, signals from the cerebral regions will be

brought into cerebellum. I would suggest that anatomical image was processed utilizing the SUIT tool (as described by Diedrichsen in 2006), which facilitated the segmentation and normalization of the cerebellum. From this segmentation, a cerebellar mask was generated, and it was manually refined as needed to make certain that no voxels from the occipital and lower temporal cortices were mistakenly incorporated. Please refer to recent cerebellum work from Jörn Diedrichsen's lab, i.e., Nettekoven, C., Zhi, D., Shahshahani, L. et al. A hierarchical atlas of the human cerebellum for functional precision mapping. *Nat Commun* 15, 8376 (2024). <https://doi.org/10.1038/s41467-024-52371-w>. The authors additionally investigated the lateralization and sexual dimorphism of cerebellar functions, which are not the highlights for the present work. I would suggest shortening them in the discussion to leave the space for the discussion of early stages of cerebello-cortical connectivity development.

Version 1:

Reviewer comments:

Reviewer #1

(Remarks to the Author)

The authors have fully addressed my concerns in this revision. I am looking forward to see the publication of this work.

Reviewer #3

(Remarks to the Author)

The authors have addressed all my concerns. Great work!

The Growing Little Brain: Cerebellar Functional Development from Cradle to School

Reviewer #1

Comment 1.0 : The study explores the functional development of the cerebellum and its connectivity with the cerebral cortex from birth to 60 months in children, utilizing over 1,000 resting-state fMRI scans. Findings reveal that the cerebellum is involved in higher-order cognitive functions early in life, highlighting a hierarchical organization and functional asymmetry between the left and right hemispheres. Notably, the research identifies sexual dimorphism in cerebellar development, indicating differing connectivity patterns between male and female children. This article provides a thorough analysis of the development of cerebellar-cortical connectivity, clearly describing the relevant findings during the neonatal phase. The structure is logical and well-organized, offering deep insights into the cerebellum's crucial role in cognitive processes, especially beyond traditional motor functions. These findings not only offer valuable insights into early childhood development but also lay the groundwork for future research in this area. However, there are still some minor questions that need further exploration to enhance the understanding of the interactions between the cerebellum and the cortex.

Response: We sincerely appreciate the reviewer's thoughtful and comprehensive summary of our study. We are grateful for the recognition of our work in exploring the functional development of the cerebellum and its connectivity with the cerebral cortex. We deeply value the reviewer's positive assessment of the manuscript, particularly the recognition of its logical structure and the insights it offers into the cerebellum's crucial role in cognitive processes, extending beyond traditional motor functions. Below, we provide detailed responses to the reviewer's comments, addressing specific questions regarding the methodology and interpretation. We are truly thankful for your constructive feedback, which we believe has significantly enhanced the clarity and robustness of our study.

Comment 1.1: While the introduction emphasizes the cerebellum's role in higher-order cognitive functions and its potential connection to neurodevelopmental disorders like ASD and ADHD, there is a noticeable gap between these claims and the actual analysis, which does not include direct cognitive or disease-specific data. Additionally, the rationale for investigating cerebellar functional gradients is not fully elaborated. Providing more context and justification for why these gradients are important for early childhood development would strengthen the overall narrative.

Response: Thank you for your insightful and constructive comments. We have carefully revised the introduction to better address the two concerns raised.

To clarify the potential contribution of our research—primarily centered on typically developing children—to the comprehension of neurodevelopmental disorders, we incorporated the following content in lines **77-84**:

“Despite these advances in understanding cerebellocortical functional connectivity in adults, research on these connections in young children is still limited. This gap in knowledge is particularly concerning given that altered cerebellocortical functional connectivity has been observed in early-onset neurodevelopmental disorders such as autism spectrum disorder (ASD) and attention deficit hyperactivity disorder (ADHD). Therefore, elucidating typical cerebellocortical functional connectivity during early development is essential for establishing a foundational understanding that can inform the interpretation of these abnormalities and their broader implications on neurodevelopmental disorders.”

This revision emphasizes the importance of characterizing typical developmental patterns as the foundation for future studies on disease-related alterations.

To better articulate the rationale for investigating cerebellar functional gradients during early childhood, we added the following content in lines **251-259**:

“Macroscale gradients of functional connectivity organize systematic information into abstract representations, providing valuable insights into how function varies across space. Using gradient-based analysis, Guell and colleagues have established the spatial macroscale gradients of the cerebellum in adults. However, the emergence and development of functional gradients in the cerebellum during childhood have not been well-explored, despite the rapid neurodevelopment occurring during this critical period. To address this gap, we employed LittleBrain to generate cerebellar functional gradient maps, complementing the functional parcellation maps by capturing subtle and gradual spatial changes in cerebellar function.”

This addition underscores the importance of functional gradients for understanding cerebellar function and provides a clearer justification for focusing on their changes during early childhood. We believe these revisions address your concerns and strengthen the overall narrative of our study.

Comment 1.2: The study makes important claims about the cerebellum’s influence on cognitive and motor outcomes, yet the results section lacks a direct examination of these outcomes. Given that the BCP dataset includes individual behavioral and cognitive measurements, the authors could strengthen their conclusions by utilizing these data. Specifically, they could test whether the cerebellar-cortical connectivity (e.g., with networks like VIS or DMN) is associated with actual cognitive and behavioral performance. Additionally, exploring whether these associations vary across different behavioral domains would provide more robust evidence to support the claim of cerebellar connectivity’s potential influence on cognitive and motor outcomes.

Response: Thank you for your insightful suggestion regarding the inclusion of behavioral and cognitive outcomes. The BCP dataset includes comprehensive and standardized Mullen Scales of Early Learning assessments for participants in early childhood, covering Gross Motor, Fine Motor, Visual Reception, Receptive Language, and Expressive Language domains. In response to your suggestion, we conducted an additional analysis examining the correlations between cerebellocortical connectivity and Mullen scores across these domains (please refer to the **Methods** section of the revised manuscript). Our results revealed robust associations between cerebellocortical connectivity and Mullen scores, for cerebellar connectivity with both primary networks and with higher-order networks. These findings suggest that cerebellocortical connectivity may be linked to specific behavioral domains, providing valuable insight into the cerebellum’s potential role in both motor and cognitive processes across early childhood. We have incorporated these results into the revised manuscript (see Table S3).

Comment 1.3: The use of a "winner-take-all" approach in the analysis may have the unintended effect of exaggerating the differences between groups. By assigning each voxel exclusively to the network with the strongest connection, this method might overlook subtler, yet meaningful, overlapping connectivity patterns and underrepresent variability within groups. This can lead to an overemphasis on the distinctions between developmental patterns (e.g. Fig 6a. volume fractions plot), potentially overstating the magnitude of sex differences (Fig. 7b).

Response: Thank you for your insightful comment on the limitations of the winner-take-all approach. We recognize that this method may overemphasize group differences by assigning each voxel exclusively to the strongest network, potentially underrepresenting overlapping connectivity patterns. To address this, we

conducted an additional analysis, calculating the volume fraction of cerebellar voxels with positive connectivity to each cortical network. This complementary approach provides a more nuanced representation of cerebellocortical connectivity, avoiding information occlusion due to overlapping patterns associated with the "winner-take-all" method. Please refer to Figure 6b and Figure S8 for these results.

We have also updated the results section as follows:

“Given that the winner-take-all approach assigns each voxel exclusively to a single network, it may fail to capture overlapping connectivity patterns that reflect functionally relevant interactions. To overcome this limitation, we also calculated the volume fraction of cerebellar voxels that show positive connectivity with each network. At coarse granularity, the volume fraction of primary networks generally declined from 0 to 12 months and remained relatively stable thereafter. In contrast, the volume fraction of higher-order networks showed a slight decline from 0 to 6 months, followed by a sustained increase. At medium granularity, the volume fractions of the SMN and AUD exhibited an overall decline from 0 to 18 months before stabilizing, whereas that of the VIS continuously decreased throughout early childhood. The volume fractions of the SN and ECN increased during the first 24 months but declined thereafter. The DMN's volume fraction decreased during the first 12 months and then increased from 12 months onward, whereas the VAN and DAN exhibited a continuous increase throughout early childhood. Notably, during the first six months of life, changes in volume fractions based on positive connectivity were relatively small compared to those observed with the winner-take-all approach. This difference occurs because the winner-take-all method prioritizes the strongest connectivity, making it more sensitive to rapid local changes in network affiliations. This effect is especially pronounced during the first six months, a critical period of rapid cerebellocortical reorganization. Therefore, these two methods offer complementary perspectives on the development of cerebellocortical connectivity.”

We appreciate the reviewer's suggestion, which has strengthened the robustness of our findings.

Comment 1.4: In the analysis of sex differences, the authors present intriguing findings regarding cerebellar connectivity, but it would be beneficial to address potential confounding variables that may influence these results. Factors such as socioeconomic status, overall head motion, and other biological variables could also impact cerebellar development and connectivity. Without controlling for these variables, the observed differences between males and females might be influenced by factors beyond sex alone. These factors should be considered or the potential impact mentioned in the discussion.

Response: We appreciate the reviewer's thoughtful comments on potential confounding variables in our analysis of sex differences in cerebellar connectivity. To address these concerns:

- **Head Motion:** During data preprocessing, we computed framewise displacement (FD) from motion parameters obtained through head motion correction. To ensure data quality, we implemented stringent quality control measures, excluding subjects with a mean Power's FD exceeding 0.5 mm. Additionally, we applied a more conservative threshold by setting Jenkinson's FD limit at 0.2 mm to further minimize the impact of motion-affected samples. Additionally, as suggested, we also performed an additional generalized additive model (GAM) analysis on the residuals from the initial GAMM fit to assess the impact of mean FD. Our results indicate no impact from wakefulness (0.00% voxels) and minimal impact from motion (<3.7% voxels). Please refer to Table S2 of the revised manuscript. Details of this analysis have been included in the **Methods** section.

- Other Confounding Variables: Unfortunately, our dataset lacked information on socioeconomic status, genetic factors, and nutritional status, so we were unable to account for these variables in our analysis. However, we acknowledge their potential influence on cerebellocortical connectivity and have added a discussion (lines **545-548**) point:

"Additionally, it is crucial to recognize that factors such as genetics, nutrition, and socioeconomic conditions may also play a role in shaping functional connectivity. Future research should account for these influences to better understand their contributions to cerebellocortical development and how they may interact with sex differences."

We have incorporated these considerations into the revised manuscript and appreciate the reviewer's insightful feedback.

Comment 1.5: The analysis of sex differences in cerebellar connectivity lacks sufficient statistical description. While differences between males and females are reported, there is limited information regarding the statistical tests used, effect sizes, or confidence intervals that would help quantify these differences.

Response: Thank you for your valuable feedback on the statistical description of sex differences in cerebellar connectivity. We obtained spatial maps of sex differences by subtracting male FC maps from female FC maps and normalizing them to a unit null distribution variance. These normalized difference maps allow us to assess statistical significance based on a normal distribution. For instance, a normalized difference of 3.1 means that females have FC values 3.1 standard deviations higher than males, corresponding to a one-tailed p-value of 0.001, indicating strong statistical significance. This approach is now summarized in the **Methods** section.

Reviewer #2

Comment 2.0: This paper provides a valuable contribution to the small but growing literature on the developing cerebellum. Given that clinical findings have shown that prenatal and early-life damage to the cerebellum can cause devastating consequences for cognitive, social, and motor functions, investigations into the developing cerebellum are greatly needed. The authors have provided a valuable roadmap for the community. The quality of the data and analyses are very high. The conclusions they draw mostly stick to findings.

Response: We sincerely thank the reviewer for their thoughtful and positive assessment of our work. We agree that investigations into the developing cerebellum are crucial, particularly given the clinical implications of early-life cerebellar damage. The reviewer's recognition of our findings as a valuable roadmap for the research community is truly encouraging. We also appreciate the reviewer's constructive suggestions regarding the use of terminology, and the statements on lateralization and sexual dimorphism. Your feedback has significantly contributed to refining our descriptions.

Comment 2.1: In the general discussion, instead of using the term "connectivity" please use "resting state connectivity" or "functional connectivity" more to remind readers that they are not looking a true, structural connectivity. As an example, you state "While there is consensus on the existence of connectivity between the cerebellum and primary networks, uncertainty remains regarding its connectivity with higher-order networks." This can be easily misconstrued; I suggest you write something like "While there is consensus on the existence of resting state connectivity between the cerebellum and primary networks (at least motor networks), uncertainty remains regarding its resting state connectivity with higher-order networks."

Response: Thank you for the valuable recommendation. We have revised the terminology across the manuscript, adopting the consistent use of "functional connectivity" to enhance clarity and mitigate any potential conflation with structural connectivity. Additionally, we have amended lines **446-448** according to your suggestion:

"While there is consensus on the existence of functional connectivity between the cerebellum and primary networks (at least motor networks), uncertainty remains regarding its functional connectivity with higher-order networks during early childhood (particularly in infancy)."

These revisions aim to enhance conceptual accuracy and improve the overall readability.

Comment 2.2: Related to my last point, you discuss visual and auditory functional connectivity. What is the ground truth for this? To my knowledge, there is nothing in optogenetics, histology, or tractography which to ground these findings. Please provide evidence or note the lack of ground truth evidence.

Response: This is a great point. As suggested, we have revised the manuscript to explicitly acknowledge the current lack of ground truth evidence from optogenetics, histology, or tractography for validating cerebellar functional connectivity with sensory regions in early childhood. Specifically, we have amended lines **433-436** to state

"Our study confirms that the developing cerebellum functionally connects with both the SMN and sensory regions such as the AUD and VIS. However, due to the lack of ground truth validation from optogenetics, histology, or tractography, these findings in early childhood remain to be fully confirmed."

Comment 2.3: The fuzziness of your terminology is found in later parts of the General Discussion, where you discuss connectivity, but in this instance, you're discussing gold standard, monkey tract-tracing. "While there is consensus on the existence of connectivity between the cerebellum and primary networks, uncertainty remains regarding its connectivity with higher-order networks." This fuzziness is acceptable in the resting state community; it is not acceptable in neuroscience literatures more connected to ground truth structural connectivity.

Response: Thank you for raising the concern about the clarity of our terminology, particularly regarding connectivity and the reference to gold-standard monkey tract-tracing studies. In response, we initially revised the manuscript to clarify the types of connectivity discussed. However, after further revisions and streamlining, we decided to remove these sections from the Discussion to better focus on the core aspect of our study—functional connectivity during early childhood. We hope this adjustment addresses your concern, and we believe the revised manuscript now offers a more focused and scientifically rigorous discussion.

Comment 2.4: There is an incorrect statement in the General Discussion "Given that language processing is predominantly lateralized to the left cortical regions and visual functions are right-lateralized..." Visual functions are NOT right lateralized. Completely wrong so should be removed.

Response: As suggested, we have now removed this statement from the manuscript. We appreciate your attention to detail and your effort in helping us improve the scientific accuracy of our work.

Comment 2.5: This statement lacks nuance to the point of being incorrect and your citations are way too old. "Sexual dimorphism in human brain structure and function is well documented." Please find a few citations from the last 10 years. No, it's not well documented in humans. There are subtle differences in some measures, but not others. Again, referring to ground-truth evidence (e.g. histology!) is the best evidence. If you don't have that evidence, clearly lay out where the evidence is from.

Response: Thank you for your valuable comment. We recognize that the statement about sexual dimorphism in human brain structure and function was overly simplistic and could have been misleading. We also agree that the concept of sexual dimorphism is not as well-documented as previously suggested, especially in terms of robust evidence. As a result, we have removed the relevant text from the manuscript. Additionally, we have replaced instances of 'sexual dimorphism' with the more precise term 'sex differences'.

Reviewer #3

Comment 3.0: Our understanding of the typical connections between the cerebellum and the cerebral cortex, especially during the earliest phases of human development, is still limited. The authors utilized functional magnetic resonance imaging (fMRI) data from over 1,000 scans of 270 children, aged from birth to 60 months, as part of the Baby Connectome Project (BCP), to chart the connectivity patterns between the cerebellum and resting-state functional networks in the brain. The study aimed to track the evolution and structure of cerebellar-cerebral connectivity, with a particular focus on when and how higher-order network connections are established. The authors employed independent component analysis (ICA) for parcellate the cerebellum functions and also utilized the most recent gradient analysis techniques to characterize the functional S-A axis. Their findings indicated that the cerebellum is connected not just to primary networks but also to higher-order networks right from birth. The detailed parcellations with high temporal resolution allowed for the capture of functional topography at various stages of development, uncovering the hierarchical arrangement of primary and higher-order networks. They also proposed an S-A axis as a key developmental trajectory for early cerebellar function. Overall, these discoveries provide new insights into the early stages of cerebellocortical connectivity development. However, some major issues should be appropriately addressed.

Response: Thank you for your thoughtful feedback on our study and for highlighting key aspects of our research. The detailed comments, particularly regarding the technical rigor of the preprocessing pipeline and the importance of ensuring accuracy in our data processing, are invaluable. We completely agree that ensuring the proper preprocessing of cerebellar data is essential for accurate cerebellocortical functional connectivity analysis, and we have carefully revised our pipeline to incorporate several improvements that ensure precise cerebellar processing. We believe these changes address your concerns and enhance the overall robustness of our study. We are grateful for the opportunity to refine our approach and provide further clarification in response to your concerns.

Comment 3.1: I noticed that imaging was carried out on children under the age of 3 (ranging from birth to 35 months) during their natural sleep, without the need for sedation. For children above the age of 3 (from 36 to 60 months), the imaging procedures were conducted either while they were sleeping or while they were passively viewing a movie. That said the brain states for children below and above the age of 3 during scanning are totally different. They are not the same condition. In other words, natural sleep and passively viewing a movie are totally different and not the “true” resting state. They can not tell us the normal function of our cerebellum. So, I’m afraid we can not take them together to characterize normal developmental trajectory for early cerebellar function.

Response: This is a great observation. Of the 1,017 scans used in our study, approximately 95% were acquired during sleep, suggesting that wakefulness has minimal impact on the results. To confirm, we conducted an additional analysis using a binary covariate to indicate wakefulness in children over 3 years old. Our results show that wakefulness had a negligible effect on our findings. Please refer to the **Methods** section and Table S2 in the revised manuscript.

Comment 3.2: I’m not convinced by the present fmri preprocessing, especially for cerebellum. The cerebellum should be separately preprocessed to guarantee the precise normalization, which is crucial to precisely parcellate the cerebellum function units and characterize the precise development trajectory of cerebellocortical connectivity in early stage. Traditional preprocessing steps are not suitable for precise cerebellum mapping. For example, regarding the Gaussian smoothing, a full-width half-maximum (FWHM) of 4mm was applied. Smoothing data for cerebral cortex and cerebellum should be separately conducted with cerebral mask or with cerebellum mask. Obviously, signals from the cerebral regions will be brought into cerebellum. I would suggest that

anatomical image was processed utilizing the SUI tool (as described by Diedrichsen in 2006), which facilitated the segmentation and normalization of the cerebellum. From this segmentation, a cerebellar mask was generated, and it was manually refined as needed to make certain that no voxels from the occipital and lower temporal cortices were mistakenly incorporated. Please refer to recent cerebellum work from Jörn Diedrichsen's lab, i.e., Nettekoven, C., Zhi, D., Shahshahani, L. et al. A hierarchical atlas of the human cerebellum for functional precision mapping. Nat Commun 15, 8376 (2024). <https://doi.org/10.1038/s41467-024-52371-w>.

Response: Thank you for your helpful suggestion regarding the preprocessing of the cerebellum and its separate treatment from the cerebrum. In response, we revised our pipeline to ensure distinct processing of cerebellar and cortical data, preventing signal leakage between the two. Specifically, we applied Gaussian smoothing separately to the cerebellum and the cerebrum. Additionally, to prevent potential signal contamination, we excluded cerebral voxels within an $8 \times 8 \times 8 \text{ mm}^3$ dilation of the cerebellar mask. These refinements enhance the accuracy of cerebellocortical functional connectivity measurements. We reprocessed the entire dataset using the revised pipeline and thoroughly reviewed our sample. These refinements enhance the robustness and reliability of our findings. Detailed information on these preprocessing steps can be found in the **Methods** section of the revised manuscript.

We appreciate the recommendation to use the SUI tool (Diedrichsen et al., 2006) for cerebellar segmentation and normalization. However, we want to note that SUI's template and segmentation methods were originally developed for adult brains (and more recently for neonatal brains). Since our dataset primarily consists of early childhood brain data, we used an infant-centric segmentation tool that is better suited to the developmental characteristics of the cerebellum in early childhood. Additionally, we have cited Nettekoven et al. (2024) in our revised manuscript, as our findings align with their proposed three-fold organization. While our study employs a different methodological approach tailored for early childhood, we acknowledge the relevance of their work in understanding cerebellar functional organization.

Comment 3.3: The authors additionally investigated the lateralization and sexual dimorphism of cerebellar functions, which are not the highlights for the present work. I would suggest shortening them in the discussion to leave the space for the discussion of early stages of cerebellocortical connectivity development.

Response: We appreciate the reviewer's suggestion regarding the discussion of lateralization and sexual dimorphism. In response, we have shortened these sections to keep the focus on the early development of cerebellocortical connectivity, allowing us to better highlight the main contributions of our study.